



# The ISLAS2020 field campaign: Studying the near-surface exchange process of stable water isotopes during the arctic wintertime

Andrew W. Seidl[1,2], Aina Johannessen[1,2], Alena Dekhtyareva[1,2], Jannis M. Huss[3,4],
Marius O. Jonassen[1,5], Alexander Schulz[6], Ove Hermansen[7], Christoph K. Thomas[3,4], and
Harald Sodemann[1,2]

[1]Geophysical Institute, University of Bergen, Bergen, Norway
[2]Bjerknes Centre for Climate Research, Bergen, Norway
[3]Micrometeorology Group, University of Bayreuth, Bayreuth, Germany
[4]Bayreuth Center of Ecology and Environmental Research, BayCEER, University of Bayreuth, Bayreuth, Germany
[5]The University Centre in Svalbard, Longyearbyen, Norway
[6]Alfred Wegener Institute, Helmholtz Centre for Polar and Marine Research, Potsdam, Germany
[7]NILU – Norwegian Institute for Air Research, Kjeller, Norway

**Correspondence:** Andrew W. Seidl (andrew.seidl@uib.no)

**Abstract.** The ISLAS2020 field campaign during February and March 2020 set out to obtain a unique dataset describing the Arctic water cycle using stable water isotope (SWI) observations. Our observation strategy focused on measuring evaporation, deposition, and precipitation, all of which are commonly sub-grid scale processes in numerical weather and climate models. Uncertain parameterizations for these processes can lead to compensating errors, which can go unnoticed; however, evaporation

and precipitation can also be investigated with SWIs, as they are an integrated tracer for processes that atmospheric moisture has undergone. The campaign can be divided into two efforts: the primary field experiment in Ny-Ålesund focused on evaporation and deposition, and the larger precipitation collection network around the Nordic Seas.

  The primary field experiment lasted three weeks, from 23 February to 15 March 2020, with temperatures reaching below -30 °C. During these weeks, we obtained near-surface, high-resolution (approx. 20 cm) SWI profiles at two deployment sites.

Using a newly developed profiling system, we measured SWI gradients in the lowermost 5 and 2 m over fjord water and snow-covered tundra, respectively. These profiles are complemented by fiber-optic distributed sensing (FODS) columns and nearby meteorological stations. The FODS columns supply continuous, high-resolution (2 cm or finer) temperature profiles above both locations, whereas the meteorological stations provide information on wind speed and direction. We also made a short deployment to the Zeppelin mountain observatory (472 m a.s.l) for measurements of the isotopic signal in the free-

troposphere. Additionally, numerous water samples from the snowpack in and around Ny-Ålesund were taken, in addition to daily fjord water samples from Kongsfjorden. These samples provide the context for the surface conditions under which profiles were collected. Isotopic connections on the synoptic scale are achieved by linking Ny-Ålesund observations with precipitation sampling at locations across the European Arctic, namely Longyearbyen, Tromsø, Andenes, Ålesund, and Bergen. The resulting dataset provides comprehensive insight into the Arctic hydrological cycle and can facilitate the study of phase

change processes and transport of water vapour into and out of the Svalbard region. Datasets from the field campaign are publicly available at the PANGAEA data repository (doi.pangaea.de/10.1594/PANGAEA.971241, Seidl et al., 2024).





# 1 Introduction

The lowermost atmosphere near the marginal ice zone is characterized by large contrasts in temperature and moisture between ice-covered and ice-free areas. In this environment, water vapour undergoes numerous phase changes, for example when evaporating from relatively warm ocean waters into overlying cold air, subsequent formation of clouds, precipitation, and the condensation of water vapour from the atmosphere onto ice and snow. Adequately representing processes from such an extreme environment in numerical models remains highly challenging (e.g. Solomon et al., 2023). Therefore, the availability of suitable measurement data from these regions is key for model verification and development (Valkonen et al., 2020).

Being a natural integrating tracer for phase changes, measurements of the stable water isotope composition in water vapour and condensed water can provide additional information when trying to understand these processes. The composition can be quantified using the $\delta$ notation (Craig, 1961) for a particular isotope, $i$, as expressed in Eq. 1.

$$\delta i = \left( \frac{{}^{i}R_{samp}}{{}^{i}R_{stnd}} - 1 \right) * 1000 \quad \permil \qquad (1)$$

where ${}^{i}R_{samp}$ is the ratio of the heavy ($i$) to light isotopologue in the measured sample, ${}^{i}R_{stnd}$ is the same in a known reference standard, and $\delta i$ is given in units of $\permil$ (parts per thousand, or permil). By studying how $\delta$ values change throughout the atmospheric water cycle, we gain knowledge on processes occurring during airmass transformation, for example the transformation from dry, polar to sub-arctic, moist airmasses, and vice versa (Thurnherr et al., 2021). Several regional models and Earth system models have been equipped with water vapour isotope physics (e.g. Brady et al., 2019; Pfahl et al., 2012). However, there is so far a severe lack of high-resolution in-situ measurements of the water isotope composition during both condensation and evaporation in Arctic conditions, as well as the conservation of the isotopic imprint during further airmass transformation over open waters. Such observational data are highly needed to test and validate different phase change processes in isotope-enabled models.

Existing theoretical work of the isotope flux during evaporation (Craig and Gordon, 1965), and available laboratory studies (Cappa, 2003; Barkan and Luz, 2007; Jouzel and Merlivat, 1984; Ellehoj et al., 2013) predict that the flux carries a specific Arctic signature, in particular for the deuterium excess (d-xs) parameter, defined as d-xs $= \delta D - 8 * \delta^{18}O$. During evaporation in non-equilibrium conditions, characterized by surface gradients in wind, temperature and water vapour mixing ratio, the HDO molecules are more likely to evaporate than the $H_2^{18}O$ molecules, a process known as kinetic isotope fractionation, which results in a positive d-excess signature. In contrast, several long-range transport studies (Klein et al., 2015; Kopec et al., 2016) have associated a negative d-excess signature to Arctic moisture. Other recent studies underscore that both positive and negative atmospheric d-excess signatures may be found in polar regions (Thurnherr et al., 2021; Brunello et al., 2023). A more nuanced view of the Arctic d-excess signal may thus be needed.

Near the ice edge, and near leads in sea ice, relatively warm open water comes into contact with the atmosphere above, and establishes neutral to unstable stratification in the surface layer, corresponding to positive (upwards) fluxes of sensible and latent heat. These positive fluxes are especially intense during marine Cold Air Outbreaks (CAOs; Papritz and Pfahl (2016)), where freeze-dried polar airmasses are transported southward over the ice edge and onto open water, resulting in a positive





d-excess signature. Conversely, in the Arctic wintertime, stable stratification frequently prevails in the surface layer over snow and ice-covered surfaces, which can lead to hoar deposition as a result of a negative (downwards) fluxes of sensible and latent heat, which could in contrast create a negative d-excess in the remaining water vapour. In the marginal ice zone (MIZ), such acutely opposed surface fluxes could occur virtually simultaneously within a very short distance.

To reconcile theory and laboratory studies of kinetic isotope fractionation during phase changes with, for example station
measurements from a fixed height above the surface, highly detailed in-situ measurements over different surface conditions are needed. In particular, to characterize the imprint of phase-change processes on the atmospheric d-excess in often very stable stratification, highly-resolved vertical profile measurements of the isotope composition and the thermodynamic environment can bridge the gap between laboratory studies and typical station-based measurements of SWI from semi-permanent installations (Leroy-Dos Santos et al., 2020; Brunello et al., 2023). Such profile measurements also complement switching manifold
installations which have been operated on several occasions at high latitudes to obtain profiles and isotope fluxes (Steen-Larsen et al., 2013; Berkelhammer et al., 2016; Wahl et al., 2021). Thereby, one gains access to both the source signature of Arctic moisture, as well essential information for subsequent transformation of this signal when transported southward away from the Arctic.

Here we present a comprehensive dataset of the water isotope composition of atmospheric water vapour, surface snow, sea
water, and precipitation from the European Arctic and downstream locations in Norway, obtained during the ISLAS2020 field campaign in Feb-Mar 2020. Using a near-surface profiling system (Seidl et al., 2023), we have performed numerous SWI profiles of the surface layer in Ny-Ålesund, Svalbard. Our deployments over snow-covered tundra and coastal fjord water probed the lowermost 2 and 5 m above the surface, respectively. Alongside these profiles, surface samples of snow and water were also collected. At both locations, in cooperation with NYTEFOX (NY-Ålesund TurbulencE Fiber Optic eXperiment) (Zeller
et al., 2021), we obtained collocated high-resolution vertical temperature measurements from fiber-optic distributed sensing (FODS) instrumentation. With the the measurement strategy employed, the FODS methods provided temperature profiles with at least 2 cm vertical resolution, down to 0.25 cm resolution. The isotopic state of the free-troposphere was obtained with a short deployment to the nearby Zeppelin mountain observatory. Throughout the campaign, we collected discrete precipitation samples in both Ny-Ålesund and Longyearbyen in Svalbard, and at four stations along the coast of the Norwegian mainland,
providing upstream and downstream context for the Arctic water isotope signatures. Our dataset thus in combination provides unprecedented water isotope information from the wintertime Arctic for use in both process studies and model development.

## 2   Campaign setting and weather conditions

The ISLAS2020 field campaign took place over a 35 day period from 20 February to 25 March 2020, with the primary experiment site being in the scientific settlement of Ny-Ålesund (78.925°N, 11.935°E), Svalbard from 23 February to 15 March. The
campaign was organized into several measurement deployments at the primary experimental site, and precipitation sampling in a corresponding network at locations downstream to the south. We first provide an overview of the primary site's general





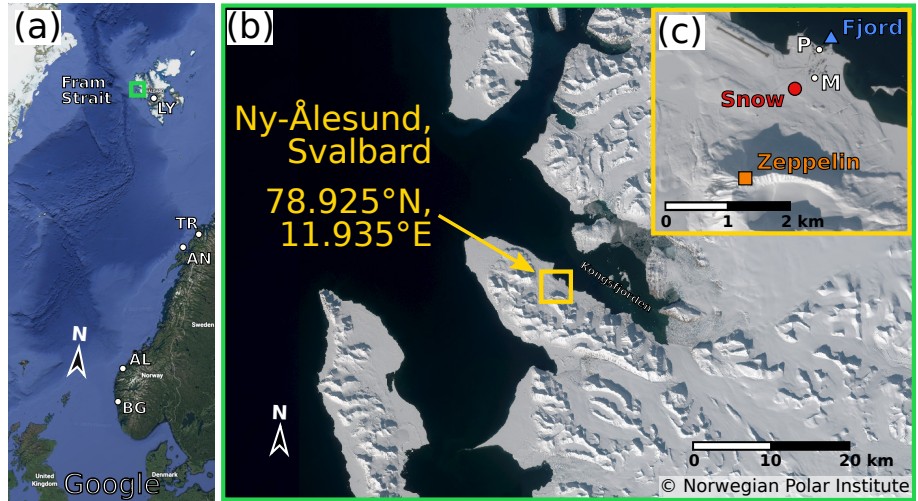

**Figure 1.** The study region. (a) Primary experiment area around Ny-Ålesund (green square) alongside additional precipitation collection sites (LY: Longyearbyen, TR: Tromsø, AN: Andenes, ÅL: Ålesund, BG: Bergen) (© Google Earth 2022). (b) The western coast of Spitsbergen and Kongsfjorden, with the settlement of Ny-Ålesund along the southern shore (yellow square). (c) The three deployment sites (Snow, Fjord, and Zeppelin) in relation to Ny-Ålesund. Also shown is the location of the precipitation collector (P) and the 10 m meteorological mast (M). (b,c: © Norwegian Polar Institute, http://toposvalbard.npolar.no).

characteristics, as well as those of the corresponding sampling network, followed by a short overview of the meteorological conditions during the campaign period.

## 2.1 Primary experiment site: Ny-Ålesund

The primary experimental site of the ISLAS2020 campaign was the scientific settlement of Ny-Ålesund, Svalbard, Norway. Ny-Ålesund is located on the southern shore of Kongsfjorden, on the western coast of Spitsbergen, Svalbard's largest island (Figure 1a,b). Given its close proximity to the Fram Strait, the location is well-suited to measure in conditions that can occur within the MIZ, for example during CAOs. Long-term measurement data from Ny-Ålesund from multiple international atmospheric research efforts thereby provide context to the observations performed in spring 2020. A 10 m meteorological mast

has been in operation near the southern edge of the settlement by the Alfred Wegener Institute (AWI) since 1993 (Maturilli et al., 2013b) (Figure 1c, "M"). The 1999-2019 February-March climatology has a median 2 m air temperature of -10.8 °C, with a median 2 m dewpoint temperature of -15.9 °C (Maturilli et al., 2013a; Maturilli, 2020). Despite the average sub-zero temperatures, since 2012 the fjord typically has less than 50 % fast-ice coverage during February and March, with a negative trend of about -4 %/year since 2003 (Gerland et al., 2020). The ice is also typically more concentrated towards the northern

shore of the fjord. The topography around the fjord influences the wind speeds observed at 10 m above ground, with the main flow direction coming from the SE, parallel with the fjord axis (Maturilli et al., 2013a; Maturilli, 2020) (Figure 1b). Almost perpendicular to the main fjord axis is the second most prevalent wind direction, coming from the WSW (Maturilli et al.,





2013a; Maturilli, 2020), with mostly slow winds outflowing from the nearby Brøgger glaciers (Schulz, 2017). Less frequently, higher wind speeds are associated with winds from the WNW that enter from Fram Strait along the main fjord axis.

We conducted detailed profiling operations of the near-surface atmosphere during different weather conditions at two deployment locations. First, profile measurements were performed over the snow-covered tundra in the vicinity of a fiber-optic distributed sensing (FODS) measurement network set up within the NYTEFOX experiment (Zeller et al., 2021), about 300 m south of the settlement (Figure 1c, site "Snow" at 78.92117°N, 11.91361°E, 27 m a.s.l.). Thereafter, profile measurements were obtained from a concrete pier over the partly ice-covered water of Kongsfjorden near the Marine Laboratory in the
northern part of the settlement (Figure 1c, site "Fjord", 78.92873°N,11.93552°E).

Between both profiling deployments, isotopic measurements of ambient vapour were performed at the Zeppelin Observatory (hereafter referred to as "Zeppelin") on Zeppelin Mountain at 472 m a.s.l and approximately 2 km to the SW of Ny-Ålesund (Figure 1c). These observations provided insight into the isotope composition of the free troposphere, giving context to the near-surface profiles measured at Ny-Ålesund.

## 2.2  Precipitation collection network

In addition to the profile measurements at Ny-Ålesund, a network for collection of discrete precipitation samples was operated at a total of six locations (Ny-Ålesund, Longyearbyen (LY), Tromsø (TR), Andenes (AN), Ålesund (AL), and Bergen (BG), Figure 1a). Longyearbyen is also in Svalbard and approximately 110 km away from Ny-Ålesund. Tromsø, Andenes, Ålesund, and Bergen are located across the Norwegian sea, on the Norwegian mainland, at a distance of 1050 km, 1100 km, 1850 km,
and 2100 km, respectively. Sample collection at all sites followed a common protocol described in Sec. 3.4.

In Ny-Ålesund, precipitation sampling was carried out at the top of the west side staircase of the Marine Laboratory building (78.92731°N, 11.93064°E) at about 5 m above ground and sea level (Figure 1c, "P"). The sampling site is sheltered to the east by the upper two stories of the Marine Laboratory. These measurements complement the continuous weekly precipitation sampling at Ny-Ålesund conducted by the Norwegian Polar Institute (Divine et al., 2011). At Ny-Ålesund, we also collected
discrete samples of the snow pack, hoar frost, sea ice, sea water for isotope analysis.

In Longyearbyen, precipitation was collected on the roof of the University Centre in Svalbard campus building (78.22262° N, 15.65251° E), approximately 12 m above the local terrain (23 m a.s.l.). Daily samples were taken, with samples consisting entirely of snow.

In Tromsø, precipitation sampling took place at three different locations. Firstly, the daily sampling location was on the roof
of the Meteorological institute, Vervarslinga for Nord-Norge (69.65384°N, 18.93672°E), at approximately 20 m above the ground (120 m a.s.l.). Secondly, higher-frequency sampling was done during weekdays at the roof of Teknologibygget of the University of Tromsø (69.68148°N, 18.97846°E) at 30 m above ground (60 m a.s.l.). Thirdly, high frequency sampling during weekends and high frequency sampling on all days after 12 March 2020 took place in a garden in Ramfjordbotn (69.52292°N, 19.24701°E) at ground level (23 m a.s.l.).

In Andenes, sample collection was done in cooperation with the Cold-Air Outbreaks in the Marine Boundary Layer Experiment (COMBLE) field campaign (Geerts et al., 2021). The campaign conducted a combination of daily and high-frequency

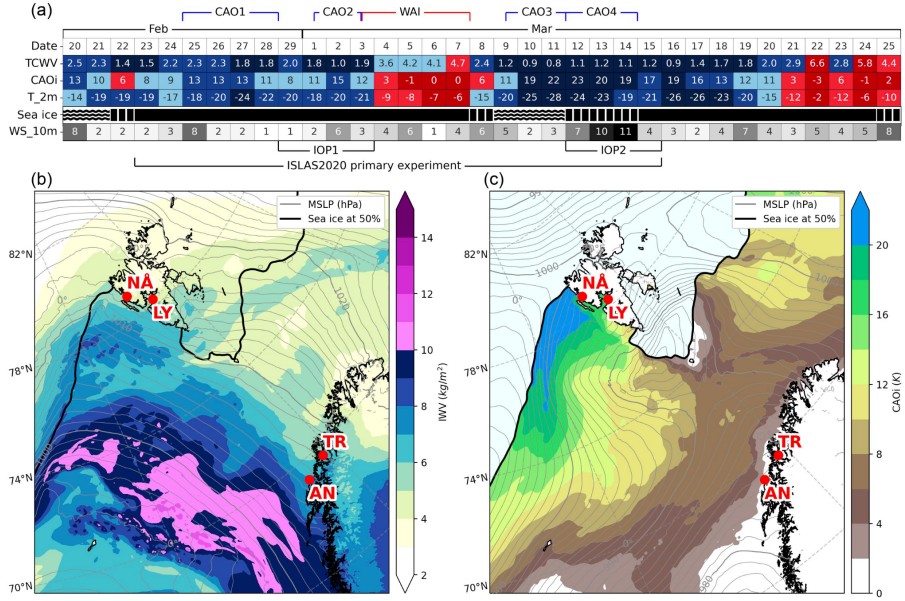

**Figure 2.** Synoptic conditions during the ISLAS2020 field campaign. (a) Overview table of conditions in Ny-Ålesund during the campaign. Total column water vapour (TCWV) $(\mathrm{kg\,m^{-2}})$ and Cold air outbreak index (CAOi) (K) from ERA5. $2\,\mathrm{m}$ Temperature and $10\,\mathrm{m}$ wind from AWI weather station. Sea-ice observation at the new pier. (b) Integrated water vapour (IWV) (shading) alongside sea level pressure (MSLP) (grey contours) for 5 March 2020 00Z from AROME-Arctic. (c) The CAOi (shading) for 11 March 2020 12Z from AROME-Arctic. Thick black line denotes sea ice concentration at $50\,\%$.

sampling beside Nordmela harbour (69.137°N, 15.672°E) at approximately $10\,\mathrm{m\,a.s.l.}$. The sampling location is described in detail in Sodemann et al. (2020).

Near Ålesund, at Barstadvik (62.36019°N, 6.26959°E) the sample collector was installed at ground level ($9\,\mathrm{m\,a.s.l.}$) in an
open area with low vegetation. Precipitation was collected on a daily basis, with additional higher-frequency sampling periods.

In Bergen, precipitation samples were collected on the roof of the Geophysical Institute (60.38368°N, 5.33191°E) at the University of Bergen approximately $30\,\mathrm{m}$ above the ground ($46\,\mathrm{m\,a.s.l.}$) until 12 March 2020. After 12 March, sample collection was moved about $2.5\,\mathrm{km}$ away to the Årstad borough of the city (60.36433°N, 5.35079°E), at ground level ($46\,\mathrm{m\,a.s.l.}$).

### 2.3 Overview of synoptic conditions during the campaign

The campaign period, lasting from 20 February to 25 March 2020 was dominated by northerly flow, with some interspersed periods of warm air advection. During the first ten days of the primary experiment (23 February to 3 March), the large-scale circulation brought cold and dry airmasses from the interior of the Arctic towards Ny-Ålesund from the north. The cold air overlying the warm, open ocean established distinct marine CAO conditions (Figure 2a). The strength of a CAO is quantified by the CAO index (CAOi) which is the difference between the potential temperature at the sea surface and $850\,\mathrm{hPa}$. These CAO



events (CAO1 and CAO2, Figure 2a) proceeded south, reaching as far south as Ålesund. Thereafter, around 3 March, a low pressure system approaching from the south reversed the flow at Ny-Ålesund. This flow reversal was associated with warmer, more moist conditions with total column water vapour (TCWV) up to $4.7 \, \text{kg} \, \text{m}^{-2}$ (Figure 2a,b). Such a weather pattern into the Arctic can be described as a Warm Air Intrusion (WAI, Dunn and Grice (1975)). The WAI lasted until about 8 March; after which, CAO conditions returned until the end of the primary experiment (Figure 2a). These CAO (CAO3 and CAO4) periods were stronger than their predecessors (CAOi > 20, Figure 2a,c), with CAO4 coming with higher wind speeds. During these periods, the entire sampling network encountered CAO conditions. These CAOs were of record strength for the Fram Strait for that time of year (Dahlke et al., 2022).

We performed profiling operations during two of the CAO periods (CAO1 and CAO3). The first was at the Snow site from 24 February 20Z until 28 February 14Z, at the southern tip of the NYTEFOX FODS array. The second was at the Fjord site from 7 March 00Z until 14 March 14Z. During the profiling deployments, the ambient temperature was almost 10 K colder (-19.7 °C) than the climatological median, and had a median dewpoint of -24.3 °C (Maturilli et al., 2013a; Maturilli, 2020). In addition, the water vapour mixing ratio at 2 m above ground was half the climatological value (median of 0.54 as compared to $1.10 \, \text{g} \, \text{kg}^{-1}$). The low temperatures during (and preceding) CAO1 also brought abnormally extensive sea-ice coverage (Figure 3a). At the end of the WAI on 8 March, the sea ice broke up and exposed open water at the Fjord site (Figure 3b).

Measurements at Zeppelin were conducted between 29 February 16Z and 3 March 08Z. The CAO2 period therefore took place within this free tropospheric observation period, including the transitory periods before and after. Additionally, two Intense Observing Periods (IOPs) were initiated during the primary experiment, involving higher frequency sample collection across the sampling network, where possible. IOP1 occurred between CAO1 and the WAI, entirely including CAO2, from 29 February to 3 March. IOP2 began on 12 March and ended on 15 March, coinciding with the onset and conclusion of CAO4.

## 3 Primary experiment site setup and operations

In order to obtain profiles of water vapour isotope composition in near-surface water vapour, specific configurations of instrumentation were deployed at sites Snow and Fjord. Though the exact configuration varied between the deployment sites, both sites required common components of instrumentation (Figure 4): (1) a cavity ring-down spectrometer (CRDS) inside a field deployment system attached to an articulating profiling arm (Seidl et al., 2023) (Figure 4, **ic**); (2) a nearby fiber-optic distributed temperature sensing column (Figure 4, **fc**); and (3) a nearby meteorological station providing information on the ambient conditions (Figure 4, **ec** and **ws**). The instrument setup at Zeppelin involved only a CRDS analyzer measuring nearby to a meteorological station.

We now present the details of the deployed installation at the three locations Snow site, Fjord site and Zeppelin. The calibration and post-processing protocols for the CRDS isotopic measurements are described jointly for all sites in Sec. 4.1.1.



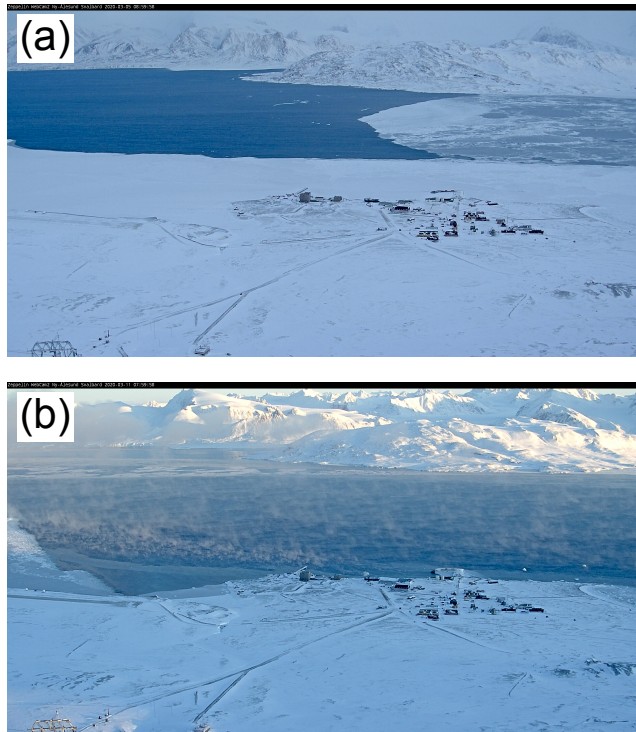

**Figure 3.** Sea ice conditions for Kongsfjorden from (a) 5 March 2020 9Z, and (b) From 11 March 2020 8Z, from the Zeppelin mountain observatory (472 m a.s.l) (Pedersen, 2013)

### 3.1 Snow deployment site setup and operations

We acquired detailed near-surface profiles of water vapour isotope composition using the articulating sampling arm of the so-called ISE-CUBE profiling system (Seidl et al., 2023), supplemented with additional instrumentation (Table 1). A Picarro L2130-i CRDS analyzer (Picarro Inc., Sunnyvale, USA; Ser#: HIDS2254) and its necessary accessories were encased in a field deployment system consisting of three ventilated, weatherproof plastic cases (Seidl et al., 2023). The analyzer is configured to measure at 4 Hz with a flow rate of around $0.15 \, \mathrm{L\,min^{-1}}$ through the measurement cavity. The sample inlet of the analyzer is connected via heated, stainless steel tubing to an articulating arm mounted to a tripod (Figure 4b,c; **ic**). The head height of this arm can be adjusted between 4 and approximately 200 cm above the surface, allowing for vapor sampling at any height in this range. Attached to the head are height and temperature sensors, logging at 1 Hz to a microcontroller (Arduino Uno). The system allows for the acquisition of vertical profiles with a temporal resolution of at least 4 min (Seidl et al., 2023). The measurements over the snow were conducted from 24 February 20:00Z to 28 February 14:15Z, and yielded six profiles with between 110 to 248 data points, corresponding to 55 to 124 min (Figure 9). A detailed example of such data is given in Sec. 5.1).

Very high-resolution profiles of the air temperature were acquired approximately 20 m to the southeast of the ISE-CUBEs by a 2.5 m tall FODS column as part of the NYTEFOX campaign (Figure 4b,c; **fc**)(Zeller et al., 2021). The corresponding dataset

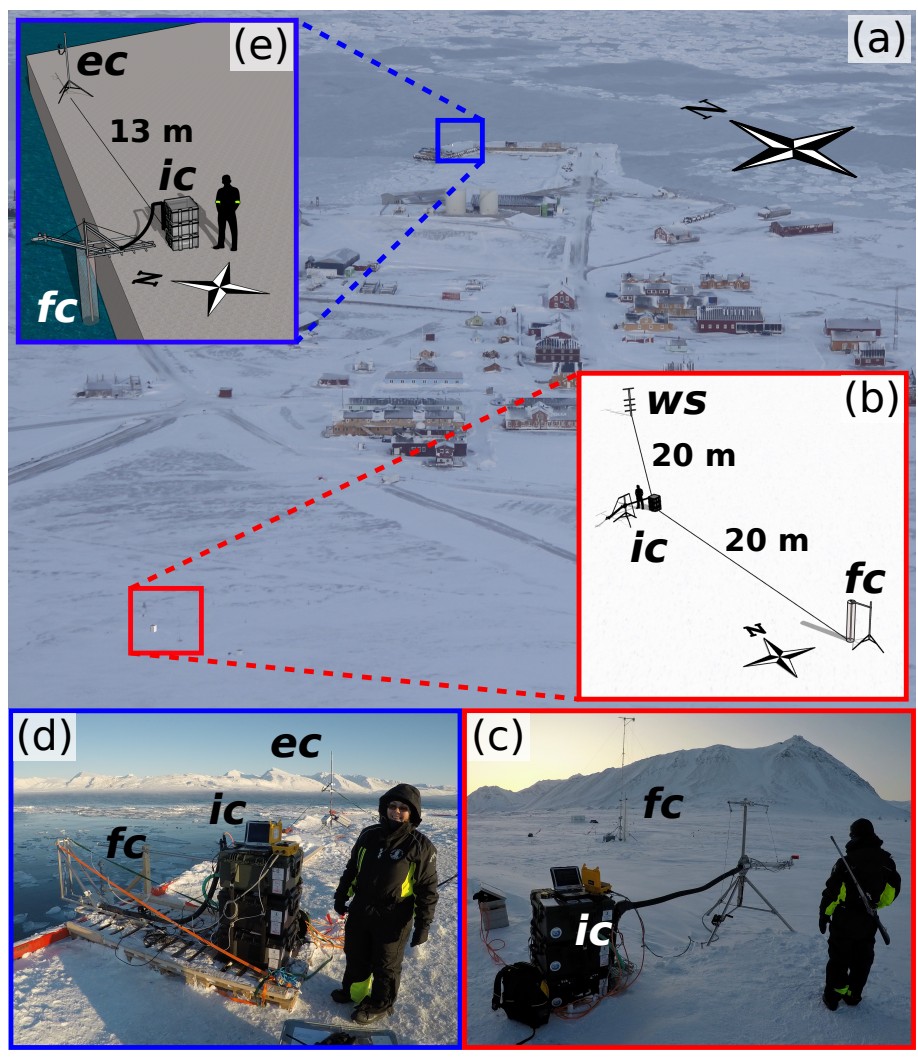

**Figure 4.** An overview of the two profiling deployment sites. (a) Ny-Ålesund as seen from Zeppelin Observatory (From 1 March 2020), including positions of Snow (red square) and Fjord (blue square) deployment sites. (b) Site schematic of the Snow site, and (c) Field photo of the Snow site, taken facing approximately south. Snow instrument abbreviations: **ic**: ICE-CUBE profiling system; **fc**: Fiber-optic distributed temperature sensing column; **ws**: automatic weather station operated by the Alfred Wegener Institute. Note that **ws** is behind the picture field of view. (d) Field photo of the Fjord site, facing northwest, and (e) Site schematic of the Fjord site. Fjord instrument abbreviations: **ic**: ICE-CUBE system; **fc**: Fiber-optic distributed temperature sensing column; **ec**: temporary eddy-covariance station.



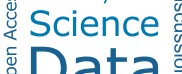

**Table 1.** Instruments deployed/present at the Snow site from 24 Feb to 28 Feb, with corresponding measurement parameters, units, and measurement height range. Heights are given relative to local ground level (approximately 27 m a.s.l., EPSG:4258).

| Instrument | Parameter | Unit | Height (m) |
|---|---|---|---|
| ISE-CUBEs | $\delta^{18}$O, $\delta$D | ‰ | 0.04 to 2.02 |
| | Specific humidity (at inlet) | g kg$^{-1}$ | 0.04 to 2.02 |
| | Air temperature (at inlet) | K | 0.04 to 2.02 |
| | Inlet height | m | 0.04 to 2.02 |
| Automatic weather station | Air temperature | K | 2.1, 1.5, 1.0, 0.5 |
| | Wind speed and direction | m s$^{-1}$ | 2.1, 1.5, 1.0, 0.5 |
| FODS column | Temperature | K | -0.25 to 2.25 |
| Snowpack samples | Surface and hoar | - | -0.01 and -0.002 |

is available at Huss et al. (2021). In this measurement setup, a fiber-optic cable was helically wrapped around a cylindrical glass-mesh frame (32 cm diameter), with vertical spacing between wraps varying with distance along the frame. With a spatial sampling resolution of 25.4 cm along the fiber itself, the column therefore had an effective vertical resolution of 0.25 cm in the lowermost quarter, and 0.5 cm, 1 cm and 2 cm in the following quarters. Plastic rings supported the glass-mesh frame at top and bottom, as well as approximately 1.2 m up the column, between the 2 and 4 cm spaced coils. These rings were then

attached to a central aluminium rod. This rod was suspended from a support arm approximately 2.4 m up a single mast tower. The bottom of the column was buried in the underlying snowpack, yielding temperature measurements across the snow-air interface. The column was in (non-continuous) operation from 26 February 13:44Z to 10 March 14:55Z (Figure 9), providing data at 0.25 to 2 cm vertical and 9 s temporal resolution with a typical precision of 0.28 °C.

Approximately 20 m north of the ISE-CUBE installation site, an automatic weather station monitored ambient conditions

(Figure 4b; **ws**). Operated by the Alfred Wegener Institute since 2010 (Jocher et al., 2012; Schulz, 2017), this station records temperature and wind conditions at four levels (0.5, 1.0, 1.5, and 2.1 m). Temperature sensors (2.1280.00.160, Adolf Thies GmbH & Co. KG) are housed in ventilated shields (1.1025.55.100, Adolf Thies GmbH & Co. KG). Wind conditions are monitored with propeller style wind anemometer/vane combinations (5106-5 at 0.5 1.0, 1.5 m and 5103-5 at 2.1 m; R. M. Young Company). The station also monitors infrared surface temperature (IR100, Campbell Scientific, Ltd.) which can be

used to derive snow surface temperature. The station measures at 0.5 Hz and records to a CR3000 datalogger (Campbell Scientific, Ltd.).

In order to capture isotopic evolution of the surface snow underlying our profile, we collected a daily sample of the uppermost layer of the surface snow from 25 to 28 February 2020 in an undisturbed section of snow approximately 3 to 5 m away from the profiling system. Surface snow samples were always taken in the afternoon, between 13:15 and 15:15Z. Using a small

pre-chilled plastic spoon, the uppermost 1 cm of a less than 50 cm$^2$ patch of snow was scraped off and filled into a plastic sample bag (58 mL Whirl-Pak, Nasco Sampling). These samples were then left to melt in the Marine Laboratory before being



**Table 2.** Discrete water samples taken during the primary field experiment for stable water isotope analysis.

| Coverage Start | Coverage End | Sample Type | Location | Count |
|---|---|---|---|---|
| 2020-02-25 | 2020-02-28 | Surface snow | Snow site | 4 |
| 2020-02-25 | 2020-02-28 | Hoar (frost flowers) | Snow site | 3 |
| 2020-02-26 | 2020-02-26 | Snow pit | Gruvebadet | 3 |
| 2020-02-23 | 2020-03-15 | Fjord water | Fjord site | 22 |
| 2020-03-01 | 2020-03-03 | Surface snow | Zeppelin | 3 |
| 2020-02-24 | 2020-03-15 | Snow (precipitation) | Marine Laboratory | 29 |

transferred to 1.5 mL glass vials (2-SVW Chromacol, Thermo Fisher Scientific Inc.) and immediately capped and stored until laboratory analysis (Sec. 4.4). A similar methodology was applied to the collection of hoar frost on the snow surface on 27 and 28 February, albeit scraping a larger area of only the uppermost 0.1 to 0.2 cm. In addition, snowpack samples were taken at 3 different layers in the nearby Gruvebadet site during the routine black carbon sampling. An overview of samples collected at the Snow site and surrounding area is given in Table 2 and Figure 10.

## 3.2 Fjord deployment site setup and operations

A second set of near-surface profiles of water vapour isotope composition was obtained at the Fjord site (Figure 1c). Fjord site measurements combined the ISE-CUBE profiling system for water vapour isotope composition, a FODS column for temperature profiles, and an eddy-covariance station for ambient meteorological parameters (Table 3). The analyzer measured again at 4 Hz with a flow rate of around $0.15\,\mathrm{L\,min^{-1}}$ through the measurement cavity. Measurement frequencies for the CRDS analyzer and profiling arm remained the same. However, due to the differing surface condition (sea ice and open water), some modifications were required to the setup. A detailed example of measurement data from the Fjord site for 10 March is given in Sec. 5.1.

Since measurements were made from the pier wall over open water or sea ice, the ISE-CUBE system with its aluminium profiling arm was mounted in a tipping frame (Figure 4d,e; **ic**). This tipping steel frame allowed reaching an additional 1.5 m below the level of the concrete pier, substantially closer to the surface of the water (Seidl et al., 2023). Given the tidal variations at the site, this extension effectively allowed for profiles between just above 0 m to approximately 5 m above water level. In order to monitor the distance above the water in this downward orientation, the profiling head contained an additional front-facing ultrasonic sensor. Using this setup, a total of 5 profiles were performed on 3 different days. Profiling ranged from 0.36 to 4.74 m above the surface and lasted between 37 to 143 min. During the remaining measurement time at the Fjord site, the inlet head was kept at or above pier level (2.5 to 5.0 m above the water) to measure the near-surface water vapour isotope composition at fixed level. This procedure also minimized risk from sea-spray entering the system, especially during the high wind conditions on 13 and 14 March (Figure 2a).



**Table 3.** Instruments deployed at the Fjord site from 7 Mar to 15 Mar, with corresponding measurement parameters, units, and measurement height range. Heights are given relative to instantaneous (fjord) water level (EPSG:5829).

| Instrument | Parameter | Unit | Height (m) |
|---|---|---|---|
| ISE-CUBEs | $\delta^{18}$O, $\delta$D | ‰ | 0.36 to 4.97 |
| | Specific humidity (at inlet) | $g\,kg^{-1}$ | 0.36 to 4.97 |
| | Air temperature (at inlet) | K | 0.36 to 4.97 |
| | Inlet height | m | 0.36 to 4.97 |
| Eddy covariance station | Air temperature (sonic) | K | 4.0 to 6.0 (2.0 above pier) |
| | Wind speed and direction | $m\,s^{-1}$ | 4.0 to 6.0 (2.0 above pier) |
| | Air pressure | hPa | 4.0 to 6.0 (2.0 above pier) |
| FODS column | Temperature | K | 0.0 to 3.8 |
| Fjord water samples | Surface | - | -0.3 to 0 |

The Fjord site FODS column was overall very similar to the Snow site column, with a diameter of 32 cm, a length of approximately 2.5 m, and with coil spacing increasing towards the upper end (see Thomas et al. (2022) for technical details). The Fjord column's central support rod was fixed to a wooden support arm at the NE edge of the ISE-CUBE wooden support platform. Then the whole column was suspended over the water, approximately 50 cm from the north face of the pier. Due to the tides, the distance between water level and column bottom fluctuated by approximately 1.5 m, including partial immersion

of the column in the fjord water; this will be discussed in Sec. 4.3. The FODS column provided temperature measurements at a vertical resolution of 0.25 to 2 cm at 0.1 Hz with a typical precision of 0.28 °C. The column began operation on 9 March at 13:52Z and provided observations until being crushed by drifting ice on 12 March at approximately 00:00Z, whereafter data quality significantly degraded.

    Ambient meteorological information at the Fjord site was acquired from an eddy-covariance station, located 13 m to the NE

of the ISE-CUBE setup (Figure 4d, **ec**). A sonic anemometer (CSAT3, Campbell Scientific Ltd.), provided 3D wind information in addition to sonic temperature at a measurement height of 2.0 m above the pier deck. Alongside the sonic anemometer, we installed an open path water/$CO_2$ sensor (LI-7500, LI-COR, Inc), measuring ambient atmospheric pressure, water vapour concentration, and $CO_2$ concentration. The measurement volume of the water/$CO_2$ sensor was on the order of 20 cm away from the sonic anemometer measurement volume, with the water/$CO_2$ sensor tilted about 30° from vertical. Due to tidal variations

and the height of the pier, this resulted in a measurement height of between 4.5 to 6.0 m above the water surface. Orientation of the anemometer towards 21° NNE (True) was such that the array axis was perpendicular to the predominant fjord-axis flow, leaving the typical flow direction unobstructed by the instrument and mast. The station logged the measurements from the sonic anemometer and the open path gas sensor at 20 Hz and recorded the incoming data stream to a CR5000 datalogger (Campbell Scientific, Ltd.). The installation was in place from 7 March 12:05Z to 14 March 19:47Z. However, heavy sea spray during the



**Table 4.** Instruments deployed/present at the Zeppelin site from 29 February to 3 March, with corresponding measurement parameters, units, and measurement height range. Heights are given relative to sea level (EPSG:4258).

| Instrument | Parameter | Unit | Height (m) |
|---|---|---|---|
| CRDS analyzer | $\delta^{18}O$, $\delta D$ | ‰ | 475 or 490 |
| | Specific humidity (at inlet) | $g\,kg^{-1}$ | 475 or 490 |
| Automatic weather station (Zeppelin observatory) | Air temperature | K | 490 |
| | Air pressure | hPa | 490 |
| | Specific humidity | $g\,kg^{-1}$ | 490 |
| | Wind speed and direction | $m\,s^{-1}$ | 490 |

night from 13 to 14 March significantly reduced the quality of the ultrasonic measurements after 13 March 22:30Z. Prior to this sea spray event, the sonic temperature was indistinguishable from both air temperature and virtual temperature at the given uncertainties, owing to the low humidities encountered. Therefore, the sonic temperature can be taken to be equivalent to air and virtual temperature. Additionally, a persistent instrument error only diagnosed after the conclusion of the field experiment, rendered the water vapour and $CO_2$ concentration measurements from the open path gas sensor of insufficient quality for use

and therefore they will not be discussed further.

Throughout the entire field experiment, we collected daily fjord water samples in the vicinity of the pier. To this end, the uppermost 10 to 30 cm of the water column were retrieved with a clean and dry 10 L plastic bucket that was lowered on a rope into the water. Immediately after collection, we measured the temperature and salinity in the bucket with a handheld datalogger (Ponsel Odeon X, Aqualabo Group) and sensor (Ponsel C4E, Aqualabo Group) combination. Instrument failure on

11 Mar prevented further in-situ measurements after this time. As a result of the variable sea ice conditions, both the exact location and sampling depth varied to some extent throughout the campaign. Fjord water samples were collected in 1.5 mL GC screwcap vials and sealed with parafilm until later laboratory analysis of the water isotope composition at FARLAB (University of Bergen, Bergen, Norway). In total, 22 daily samples were collected from 23 Feb to 15 Mar (Table 2 and Figure 10).

### 3.3 Zeppelin deployment site setup and operations

The deployment at the Zeppelin site consisted of the CRDS analyzer only, with corresponding calibration equipment, surface snow sampling, and additional measurement parameters acquired from permanent installation (Table 4). While located at the observatory, the analyzer sampling can be divided into three distinct periods, based on the permanent or temporary inlet line used (Table 5). From 15:30Z on 29 February to 14:40Z on 1 March (ZEP1 period), the analyzer was connected to Zeppelin's main heated inlet line (21 m, 30 °C, 10 L min$^{-1}$) via a 10 m long section of ¼-inch plastic tubing. The plastic tubing was a





**Table 5.** Periods of the deployment at the Zeppelin site from 29 February to 3 March.

| Period name | Start time (UTC) | End time (UTC) | Inlet details | Total length (m) |
|---|---|---|---|---|
| ZEP1 | 29 Feb 15:30 | 1 Mar 14:40 | Main inlet ($10\,\mathrm{L\,min^{-1}}$) and 10 m plastic ($< 0.1\,\mathrm{L\,min^{-1}}$) | 31 |
| ZEP2 | 1 Mar 15:25 | 2 Mar 14:30 | 4 m Stainless steel/PTFE, (half heated) ($9\,\mathrm{L\,min^{-1}}$) | 4 |
| ZEP3 | 2 Mar 15:20 | 3 Mar 8:30 | Main inlet ($10\,\mathrm{L\,min^{-1}}$) and 5 m PTFE ($9\,\mathrm{L\,min^{-1}}$) | 26 |

combination of Bev-A-Line (about 4 m) and PTFE tubing (about 6 m). The flowrate through the plastic tubing was equal to the flowrate through the analyzer (around $0.03\,\mathrm{L\,min^{-1}}$), as well as an identical analyzer on the same line, that is installed semi-permanently at the station. From 1 March 15:25Z until 2 March 14:30Z (ZEP2 period), we utilized a 2 m section of ¼-inch stainless steel tubing, that was heated to $60\,^{\circ}\mathrm{C}$, and that extended outside through a cable passthrough to a downward-facing inlet above the rooftop protected by a plastic cover. An additional section of 2 m ¼-inch PTFE tubing (unheated) connected

the analyzer to the stainless steel section. Finally, from 2 March 15:20Z until 3 March 8:30Z (ZEP3 period), the analyzer was connected directly to Zeppelin's main inlet with a 5 m section of ¼-inch PTFE tubing, flushed at approximately $9\,\mathrm{L\,min^{-1}}$. The analyzer also utilized a higher flowrate during the ZEP3 period, similar to the flowrate used at Snow and Fjord ($0.15\,\mathrm{L\,min^{-1}}$). Regular calibrations with the SDM module and Vapourizer were performed during ZEP1 and ZEP2 (Sec. 4.1.1).

  Ambient meteorology during the deployment was measured by Zeppelin's permanent automatic weather station, located at

approximately 15 m above the main station building. The weather station provides air temperature and humidity (HMP-155, Vaisala), air pressure (PTB210, Vaisala), and wind speed and direction (WMT700, Vaisala) at a 1 min frequency.

  During our routine operations at the observatory, we collected daily samples of the snowpack close to Zeppelin, with the exception of 29 February (Table 2). This sampling was done with a methodology similar to the snowpack collection described in Sec. 3.1.

## 3.4   Discrete sampling of precipitation

Collection of precipitation samples was conducted at the principal experimental sites and at the sites of the precipitation sampling network. Since protocols were kept consistent across the different sites, the sampling procedures described here apply to all sites.

  Precipitation samples for daily and high-frequency sampling were collected in a clear polypropylene box (approximately

60x40x40 cm LxWxH). Plastic boxes were inspected for precipitation accumulation at regular intervals, at least daily. When heavier precipitation was forecasted for particular sites, sampling frequency was increased if possible. Higher-frequency sampling periods were termed intensive observational periods (IOPs). In total, 2 IOPs were conducted, yielding higher-frequency sampling at Ny-Ålesund, Tromsø, Andenes, and Ålesund (Figure 10). Upon collection end of daily or IOP sampling, the active collection container was swapped with an identical clean and dry replacement container to minimise gaps in the sampling

period and prevent carry-over of sample material. In the case of solid precipitation, sample material was transferred to a plastic sample bag (58 mL Whirl-Pak, Nasco Sampling) with a clean and dry plastic spoon. If the volume of snow exceeded bag size,





the box contents were homogenized and an aliquot (i.e. small representative sample) was collected. After melting, the liquid water was poured into an 8.0 mL glass vial (548-0821, VWR International, LLC.) and immediately capped with rubber seal caps. In case of liquid or mixed-phase precipitation, the same plastic container and sampling protocol was used for collection, except that the liquid water would be directly transferred to an 8.0 mL glass vial. For the majority of samples, collecting personnel at all sites noted the weather conditions at time of collection, alongside temperature and wind speed, if available. A summary of precipitation samples collected in Ny-Ålesund during the primary experiment is given in Table 2, with overviews of other collection locations found in Figure 10.

## 4 Data processing, uncertainties and dataset description

### 4.1 ISE-CUBE data processing

As described in Seidl et al. (2023), the ISE-CUBEs produce two independently logged data streams, one from the CRDS analyzer containing the water isotope data, and another from the profiling arm.

#### 4.1.1 CRDS analyzer data processing

Correction and calibration of the raw measurement signal from the ISE-CUBE system is required to obtain reliable data referencing to the VSMOW-SLAP scale (IAEA, 2009). Calibrations for the Snow and Fjord deployments were performed immediately before and after each deployment of the profiling system at the Marine Laboratory in Ny-Ålesund. The secondary standards DI ($\delta^{18}$O $= -7.68 \pm 0.07\,‰$) and ($\delta$D $= -49.7 \pm 0.4\,‰$) and GSM1 ($\delta^{18}$O $= -32.90 \pm 0.05\,‰$) and ($\delta$D $= -261.6 \pm 0.3\,‰$), in use at FARLAB at the University of Bergen, Bergen, Norway, were utilized for calibration. These standards were provided to the analyzer with a Picarro Standard Delivery Module (SDM) (A0101, Picarro Inc., USA) and the Vaporizer module (A0211, Picarro Inc., USA), while a Drierite cartridge (MT400, Agilent Technologies Inc., USA) provided dry air. A detailed account of the calibrations preceding and following the Snow and Fjord deployments can be found in Seidl et al. (2023).

During the deployment at Zeppelin, the SDM/Vaporizer combination was used again with the same standards. Thereby, calibrations were conducted every 9 hours between 29 February and 2 March. Each calibration cycle lasted around 1 hour, and consisted of two segments of 20–25 min duration for each standard. A segment was considered valid for calibration if the exposure contained at least 10 consecutive minutes with humidity variation below 0.3 g/kg (500 ppmv). As was also the case for the calibrations performed at the Marine Laboratory, the standard deviation of the mean isotopic values across the 5 calibrations was smaller than the isotopic standard deviation experienced during any individual exposure, for both $\delta^{18}$O and $\delta$D. Furthermore, the calibrations at Zeppelin and for the entire campaign exhibited negligible drift (less than 0.01 ($\delta^{18}$O) and 0.02 ($\delta$D)‰ per day, see Seidl et al. (2023)).

Calibration of the water vapour isotope measurements were also obtained using the long-term slope and offset of the analyzer's raw $\delta^{18}$O and $\delta$D signals compared to secondary standards calibrated at FARLAB onto the VSMOW-SLAP scale.



Previous deployments have shown that the type of CRDS analyzers used here have a long-term drift that is similar to the measurement precision (Bailey et al., 2023). We observe this also to be the case for the specific analyzer used here (Ser. No.
HIDS2254) over a time frame of several years. In the selection of valid calibration runs, calibration segments of typically 20 min from an SDM were retained if the mixing ratio was between 18'000 and 30'000 ppmv, while low-quality calibration runs with oscillations of the water concentration due to bursting air bubbles were excluded. Calculated from more than 1000 calibrations with different secondary laboratory standards performed in the period from 20 Nov 2016 to 30 May 2022 in laboratory application and field deployments, we find mean slope and offset values of:

$$\delta^{18}O_{cal} = 1.1169 \times \delta^{18}O_{raw} + 0.7742 \quad ‰$$

$$\delta D_{cal} = 0.9163 \times \delta D_{raw} - 8.3058 \quad ‰$$

These calibration coefficients have been obtained from a linear regression to the long-term average values of 6 secondary standards calibrated to the VSMOW-SLAP scale. From the standard deviation about the regression, the calibration uncertainty
for a sample at -10 ‰ $\delta^{18}O$ and -100 ‰ is thereby estimated as 0.44 ‰ and 1.5 ‰, respectively. These long-term calibration values only shifted markedly when the analyzer was returned from service at the manufacturer after June 2022. Calibrated water vapour isotope data from this approach are denoted by appending the suffix "_LT" to the respective data variables (see Appendix B).

During the ISLAS2020 field deployment, consistency with the long-term calibration coefficients was checked from SDM
calibrations performed in a laboratory environment. The median absolute difference between the two calibration methods for the entire experiment is 0.10 ‰ for the $\delta^{18}O$ and approximately 1.0 ‰ for $\delta D$. Uncertainty has possibly been introduced in these field calibrations due to remnant humidity in the background air that was dried with a moisture trap.

For both calibration methods, the accuracy of CRDS isotopic measurements is subject to a dependency on both isotope ratio and mixing ratio (Aemisegger et al., 2012; Bonne et al., 2014; Weng et al., 2020). Since the response is often particularly
pronounced at low mixing ratios, it is crucially important for the ISLAS2020 field deployments to perform correction for this dependency. Response characteristics are different for each specific analyzer, and can be determined from providing a vapor stream of a known isotopic composition across a wide mixing ratio span. For the particular CRDS analyzer used in ISLAS2020, such a characterization was performed first in 2018 using autosampler injections of laboratory standards (Weng et al., 2020). The characterization has since been repeated several times and updated with a wider range of standards. Here we use the
correction method based on a large number of suitable characterisation measurements, involving a calibration device based on microdrop dispensing technology (Sodemann et al., 2023a). The correction for the ISLAS2020 campaign (Appendix A) is largest for low mixing ratios that are more depleted (less than -247.9 ‰ $\delta D$ and -27.95 ‰ $\delta 18O$ for mixing ratios below $0.6\,g\,kg^{-1}$). 38.6 % of the measurement dataset fall in this range. The median correction for this subset is -10.1 ‰ for $\delta D$ and 1.01 ‰ for $\delta 18O$, while the median correction for the entire dataset is -6.5 ‰ for $\delta D$ and 0.60 ‰ for $\delta 18O$. In particular
at the lowest depletion and mixing ratios, the correction has a significant impact on the d-excess, and introduces uncertainty. Estimates from a bootstrap method give an uncertainty of 0.4 ‰ for $\delta D$ and 0.1 ‰ for $\delta 18O$ for this correction (Sodemann et al., 2023a).





Variables relating to CRDS measurements can be found in Appendix B for each deployment under the "CRDS analyzer" grouping. CRDS analyzer variables are available at a 30 s resolution in the data files from Snow and Fjord, whereas Zeppelin

resolution is 2 min.

### 4.1.2 Profiling arm data processing

The profiling arm monitored and logged the ambient temperature just beside the sampling inlet, as well as the height of the inlet above the underlying surface. A temperature offset of $+0.67\,°C$ was identified from laboratory testing and accounted for in the final data set. Distance sensors onboard the profiling arm were also routinely compared with manual measurements and

found to be accurate to within 2 cm. Full details of the data processing can be found in Seidl et al. (2023). Variables relating to the profiling arm measurements can be found in Appendix B for the Snow and Fjord deployments under the "Profiling module" grouping. Similarly to the CRDS analyzer variables, profiling module variables are available at 30 s resolution.

### 4.2 Ambient meteorological measurement processing

The automatic weather station near the Snow site measured horizontal wind speed and direction, air temperature, and surface

temperature. The minimum speed for the propeller anemometers of the AWS is $1.0\,\mathrm{m\,s^{-1}}$ and $1.1\,\mathrm{m\,s^{-1}}$ for determining wind direction, resulting in increased uncertainty of winds below these thresholds. The surface temperature ($T_s$) is calculated from the Stefan-Boltzmann law from radiative flux measurements using a surface emissivity $\epsilon = 0.985$ which is typical for snow. Changing $\epsilon$ by 0.005 results in variations of approximately 0.3 K The data from the AWS is associated with the variables listed under the "Meteorological station (S)now" of Appendix B, and are available at 30 s resolution.

The eddy-covariance station installed at the Fjord site was primarily utilized to provide both the horizontal wind and air temperature at the measurement site. Wind and temperature data was quality controlled according to the internal diagnostic flagging of the sonic anemometer. Converting measured wind speeds from the instrument's local coordinate system into geographic coordinates included correcting external offset from true north (21° NNE). Atmospheric pressure measurements from the open path gas sensor required no additional processing. Once quality controlled and direction corrected, the 20 Hz signal

from the station was aggregated to 30 s averages to match with the CRDS and profiling module measurements. Wind, temperature, and pressure measurements from the eddy-covariance station are connected to the variables in the "Meteorological station (F)jord" category of Appendix B.

The Zeppelin weather station provided air temperature, relative humidity, atmospheric pressure, and horizontal wind speed and direction with associated quality scores for each. For the entire deployment period, all but the wind variables constantly

produced data of maximum quality (score 100). Wind measurements had a few brief episodes of lower quality, but still ranked a mean quality of 99.9 for the deployment period. These quality scores, in addition to the parameters themselves, are registered to the variables listed under "Meteorological station (Z)eppelin" in Appendix B. Zeppelin weather station data is available at 2 min resolution.

Decomposition of wind speed and direction into northward and eastward components, as well as the conversion from relative

to specific humidity was done via the open-source Python software package, MetPy (May et al., 2022).


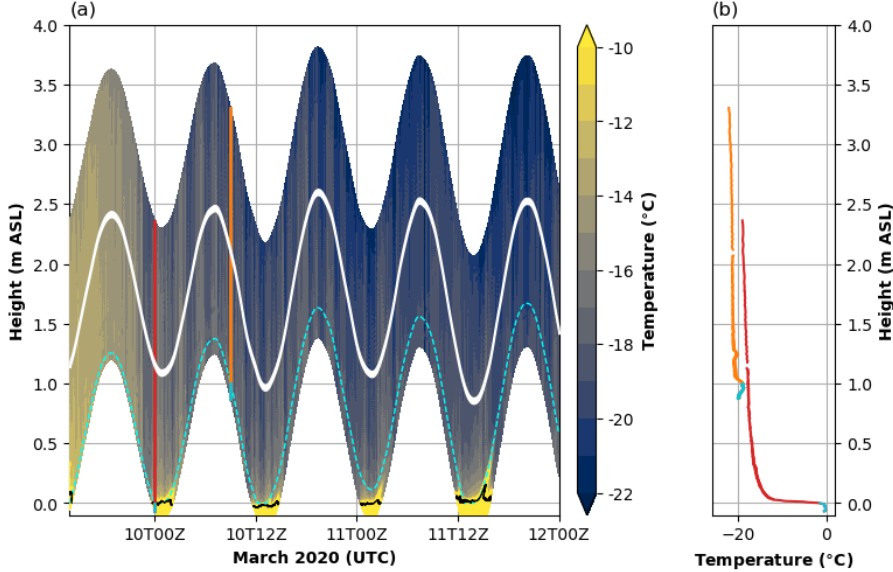

**Figure 5.** Total timeseries of the temperature observed by the Fjord FODS column (a, shading). White gap in middle of column (a,b) caused by plastic support ring. Black contour (a) is the -1.8 °C threshold, representing the water surface. Dotted cyan line (a) denotes level at which the column has been previously fully immersed in fjord water, and therefore might be prone to icing effects; this does not include the potential icing level caused by wave action, which might extend the effects further up. Red/cyan line (a,b) indicates 10 March 2020 00:00Z profile, with cyan section showing extent that the column has been previously fully immersed. Orange/gray line (a,b) is the same for 10 March 2020 19:40Z.

## 4.3 Fiber-optic distributed sensing column processing

The FODS instrument for the Snow column recorded temporal averages of 3 s. Since the instrument consecutively sampled three different fibers, samples have an effective temporal resolution of 9 s. For the Fjord FODS column temporal averages of 10 s were recorded at a rate of 10 s; further details on accuracy and bias can be found in Zeller et al. (2021) and Thomas et al.

(2022). Signal handling and datalogging at the Snow site utilized an ULTIMA DTS, 5 km variant (Silixa, UK), whereas the Fjord deployment had an XT DTS (Silixa, UK). These instruments resolve temperature measurements along the fiber-optic cable about every 0.6 m. As the circumference of the glass-mesh support is approximately 1 m, transforming this signal into a vertical profile involves converting the length along the fiber into a number of wraps around the glass-mesh according to the spacings given in Sec. 3.1. Each height level was then averaged over 30 s intervals, producing a dataset with the same time

resolution as the ISE-CUBE data.

The surface properties of the two sites required different treatment at the lowest heights of the column data. At the Snow site, the column was partly in the snowpack. The average snow height above the bottom of the column was 0.17 m during that time period, which is subsequently used as the "zero-level" in the dataset. Averaging over a band ±0.005 m on either side of





the zero-level gives the temperature in the surface snow (in the dataset as "Tsnow"), a bulk measure of the temperature across
the snow-air interface.

At the Fjord site, due to the tidal variations, determining a zero-level for the water's surface was more involved. At times
during the deployment, the bottom-most 0.25 m was immersed in the fjord water. The timing of this immersion matches with
the semi-diurnal high tide mark. We utilized the observed tidal height for Ny-Ålesund from Kartverket[1], interpolated to a
30 s resolution. Using a -1.8 °C temperature threshold (Figure 5a, black contour) as a guide for surface level, we qualitatively
matched the immersed signal with the tidal signal. This allowed to calculate the time-dependent distance between the surface
of the water and the bottom of the column, and the resulting temperature gradient (Figure 5a, shading). Owing to the immersion
capability of the FODS column, we can clearly distinguish the surface of the water at 00Z on 10 March (Figure 5, red/cyan
line). However, the immersion also led to ice buildup at air temperatures below freezing, which is evident in the temperature
anomaly in the lower levels at 09Z (Figure 5, orange/cyan line).

Data from the FODS columns from the Snow and Fjord deployments are listed with the variables found in Appendix B under
the "FODS column" grouping.

### 4.4 Laboratory analysis of liquid samples

In total, 202 samples of freshwater (rain, snow and surface snow) and seawater were collected during the ISLAS2020 campaign, at both the primary field experiment and at the downstream precipitation collection sites. All collected samples were
analysed within weeks after collection at the Facility for Advanced Isotopic Research and Monitoring of Weather, Climate and
Biogeochemical Cycling (FARLAB) at the University of Bergen. The analysis was done by liquid injections on two Picarro
L-2140i CRDS analyzers (Picarro Inc, Sunnyvale, USA; Ser#: HKDS2038 and HKDS2039), utilizing Picarro Autosamplers
(A0340) and Vaporizers (A0211). Secondary laboratory standards (Table 6) were thereby used to account for instrument drift,
as well as for calibration of samples onto the VSMOW-SLAP reference scale using the analytical sequence described in the Appendix of Sodemann et al. (2023b). The secondary standards GLW and EVAP2 were utilized for freshwater sample calibration,
with DI2 and FIN serving as drift standards. EVAP2 and DI2 were used for saltwater calibration with BERM used as a drift
standard (Table 6). The software package FLIIMP (FARLAB liquid injection isotope measurements processor) V2.0 was used
for corrections and calibration of all samples. Combined uncertainty for each sample is inlcuded in the dataset, and comprises
the calibration uncertainty of the secondary standards, the repeatability for each sample, and the long-term reproducibility of
the measurement system.

### 5 Data examples

We now present examples of profiling operations from the field experiment, as well as the time-series of precipitation samples
from across the collection network. We begin with two examples of profiles from the Snow and Fjord sites.

---

[1]https://www.kartverket.no/en/at-sea/se-havniva/result?id=1082348, last accessed: 22 December 2022



**Table 6.** Secondary lab standards used for the calibration of liquid samples onto the VSMOW-SLAP scale.

| Standard name | $\delta^{18}$O (‰) | $\delta$D (‰) |
|---|---|---|
| GLW | -40.02±0.07 | -307.79±0.75 |
| FIN | -11.60±0.10 | -80.78±0.47 |
| DI2 | -7.63±0.10 | -50.72±0.27 |
| BERM | 0.62±0.12 | 6.80±0.51 |
| EVAP2 | 1.81±0.13 | 9.52±0.65 |

## 5.1 Profiling sites

The example from the Snow site spans 4.5 h on 28 Feb 2020 (Figure 6). The temporal variation of the inlet height (Figure 6a, black line with red highlight) is set against the backdrop of the vertical temperature profile from the FODS column (shading). The snow temperature (blue line) and surface temperature (cyan line) were always several degrees below the 2.1 m ambient temperature (orange line), reflecting the very stable stratification typical of the Snow site (Figure 6b). The temperatures at the inlet height extracted from the FODS column (red line) and measured at the sensor head (black line) clearly co-vary (r=0.942),

but with a positive bias for the inlet temperature sensor of 1.5 °C. The example profile shown was made under light wind conditions with wind speeds under 3 m/s (Figure 6c, orange line). Winds were predominantly from the south (black dots), indicating air descending from the direction of the nearby inland glaciers (Figure 1c). These meteorological observations together set the context for the water vapour isotope measurements (Figure 6d). This time period was characterized by highly depleted conditions in $\delta$D reaching below -370‰ (black line). The d-excess closely followed the time evolution of the $\delta$D,

albeit at and around very negative values of -40‰. The co-variation between $\delta$D and d-excess indicates that the $\delta^{18}$O has a different variability, which is substantially smaller than $\delta$D over the different heights of the profile (not shown).

The example from the Fjord case covers 5.5 h during 10 Mar 2020 (Figure 7). In contrast to the Snow site, the temperature gradients observed by the FODS column over water are characterized by warm air close to the water surface, overlaid by colder air (Figure 7a). There are clear co-variations between air temperature measurements from the inlet (black line), from

the FODS column at inlet height (red line), and from the sonic anemometer (orange line) (Figure 7b). At the Fjord site, ambient temperatures were almost always (97.9 % of the deployment) colder than the inlet temperature, with most of the remaining time occurring during higher windspeeds (> 5 m s$^{-1}$) than those typically encountered during profiling. For the profiling period shown, wind speeds were relatively low (about 2 m s$^{-1}$) and with mainly southerly directions (Figure 7c). The corresponding measurements in isotope parameters show pronounced variation with height (Figure 7d). When the inlet was at

3.5 m above the sea surface or higher, the $\delta$D reached below -280‰ (before 9:10Z and around 10:45Z), while at lower levels the $\delta$D was substantially less depleted (about -200‰). The d-excess (black line) follows again closely along the pattern of the $\delta$D, yet at this location it exhibits predominantly positive values, except for the periods where the inlet was above 3.5 m height. We interpret this strong vertical gradient in the lowermost metres of the boundary layer as a signature of two very different airmasses meeting at the Fjord site, where the upper layer begins to resemble the conditions at the Snow site, and

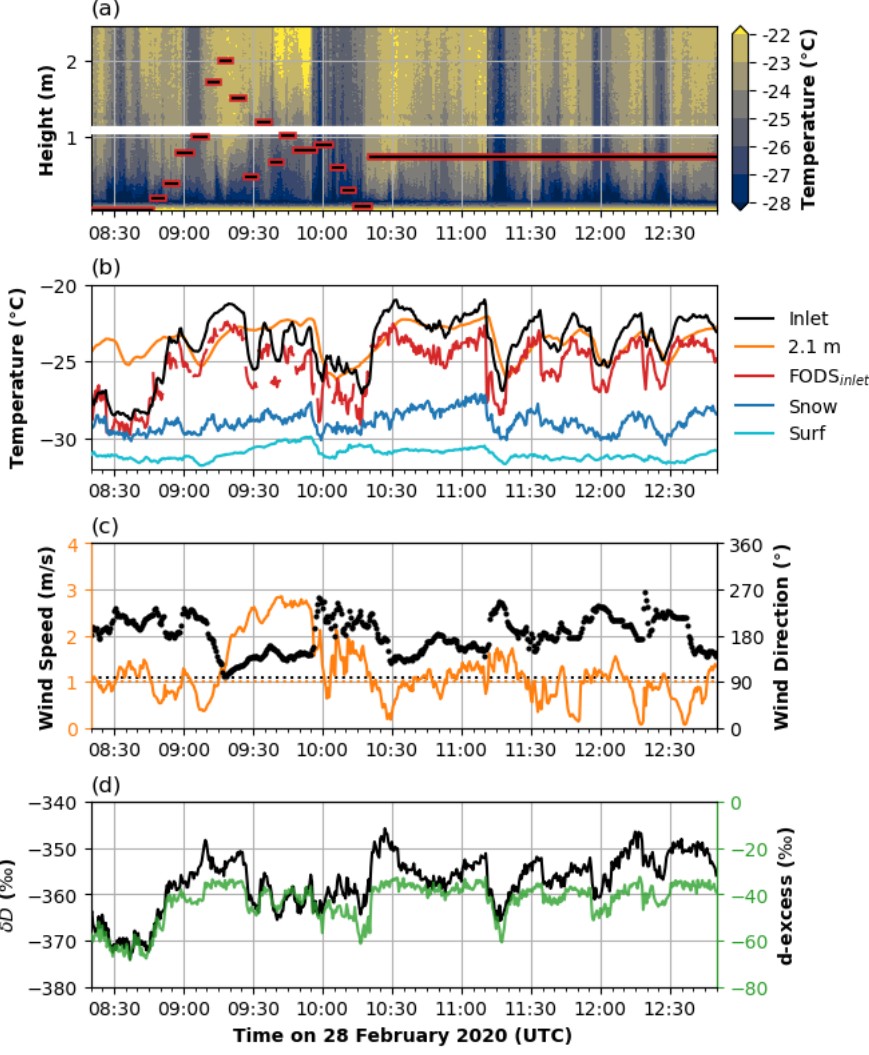

**Figure 6.** A profiling period from 28 Feb 2020 8:20 to 12:50Z over snow-covered tundra. (a) Height of sampling inlet (black/red line) displayed alongside temperatures observed with FODS column (shading). White gap caused by plastic support ring. (b) Inlet temperature (black), 2.1 m temperature (orange), and temperature of FODS column at inlet height (red), as well as snow (blue) and surface (cyan) temperature. (c) Wind speed (left axis, orange) and wind direction (right axis, black). Dotted lines indicate manufacturer's detection thresholds (i.e. minimum wind speed) for wind speed (dotted red) and direction (dotted black). (d) $\delta$D (left axis, black) and the d-excess (right axis, green).

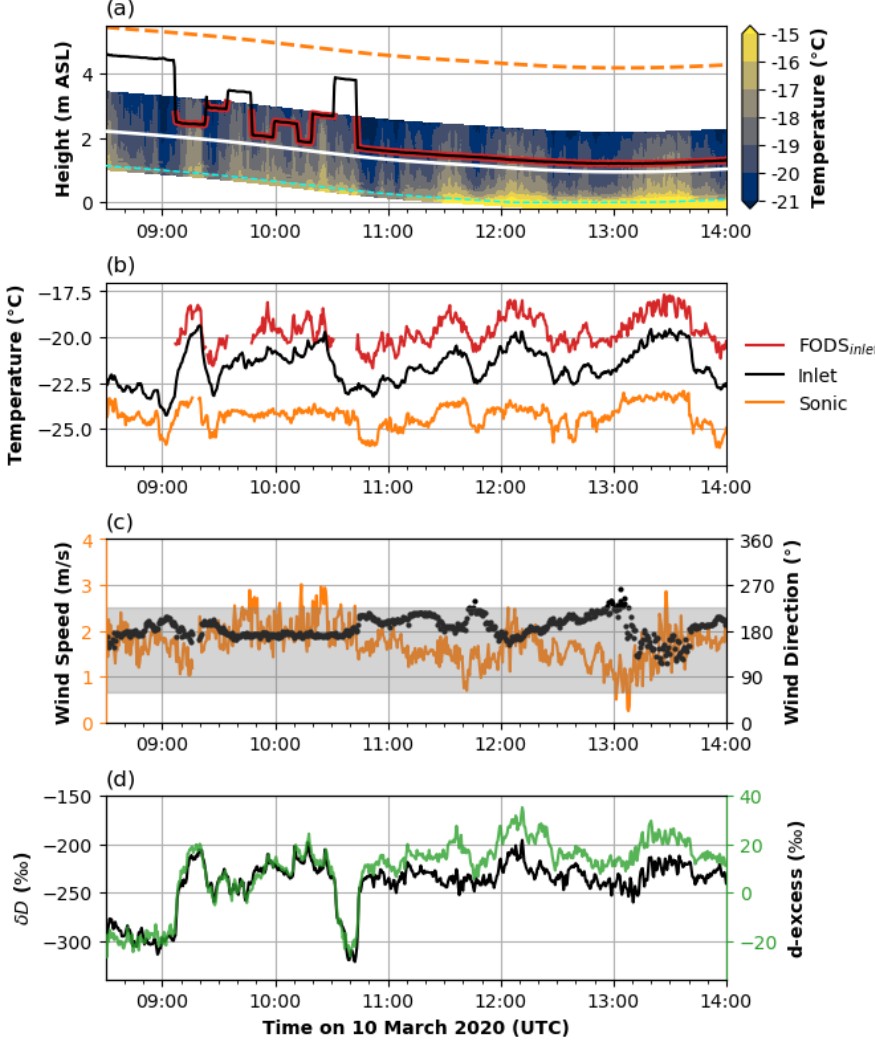

**Figure 7.** A profiling period from 10 Mar 2020 8:30 to 14:00Z over open fjord water. (a) Height of sampling inlet (black/red line) displayed alongside temperatures observed with FODS column (shading). White gap caused by plastic support ring. Dashed cyan line indicates previous/current immersion level, similar to Figure 5a. Dashed orange line is the height of the sonic anemometer, above sea level. (b) Inlet temperature (black), sonic temperature (orange), and temperature of FODS column at inlet height (red). (c) Wind speed (left axis, orange) and wind direction (right axis, black). Grey shading indicates the sector where the inlet is in the wind shadow of the concrete pier that it is deployed upon. (d) δD (left axis, black) and the d-excess (right axis, green).





**Table 7.** Statistics for isotopic composition of collected precipitation across the sampling network locations.

| Sampling location | Distance to Ny-Ålesund (km) | $\delta D_{mean} \pm \delta D_{stdev}$ [$\delta D_{min}$ , $\delta D_{max}$] (‰) | d-xs$_{mean}$ ± d-xs$_{stdev}$ [d-xs$_{min}$, d-xs$_{max}$] (‰) |
|---|---|---|---|
| Ny-Ålesund | 0 | -130.6 ± 64.0 [-282.5, -46.8] | 29.1 ± 13.0 [4.1, 45.0] |
| Longyearbyen | 110 | -107.1 ± 33.8 [-177.8, -61.3] | 29.4 ± 12.5 [4.4, 53.5] |
| Tromsø | 1050 | -83.3 ± 44.5 [-229.7, -22.7] | 20.1 ± 10.4 [-4.9, 39.2] |
| Andenes | 1100 | -58.0 ± 27.6 [-118.2, -0.1] | 19.5 ± 9.0 [-1.3, 33.0] |
| Ålesund | 1850 | -50.6 ± 28.1 [-117.8, 0.4] | 13.3 ± 12.9 [-21.9, 33.67] |
| Bergen | 2100 | -55.7 ± 24.8 [-111.0, -16.8] | 12.9 ± 3.8 [5.0, 17.6] |

the lower layer is influenced by surface evaporation. Further analysis of these differences can provide insight into the isotopic fractionation occurring between atmosphere and surface, namely during evaporation from, and deposition onto, the surface.

## 5.2 Precipitation collection

Figure 8 shows the time series of the isotopic composition of precipitation collected during the campaign, from across the sampling network. The distance of the sampling location from Ny-Ålesund increases from the top to bottom panel. Grey shading indicates the two IOP periods of the campaign. There are variations in both the $\delta D$ (black lines) and d-excess (green lines) that vary with the prevailing weather conditions, namely the WAI and different CAOs (Figure 2). More northerly locations tend to have more variable water isotope composition, with Ny-Ålesund (Figure 8a) exhibiting the largest standard deviation ($\delta D_{stdev}$=64.0‰, d-xs$_{stdev}$= 13.0‰), and Bergen (Figure 8f) the lowest ($\delta D_{stdev}$=24.8‰, d-xs$_{stdev}$= 3.8‰). Trends also appear in the mean isotopic depletion, which becomes more negative in the poleward direction, and the d-excess values, which become more positive. A statistical overview of the precipitation isotope composition from the sampling network can be found in Table 7.

## 6 Data availability

Data from the ISLAS2020 campaign can be accessed at the PANGAEA data repository at doi.org/10.1594/PANGAEA.971241 (Seidl et al., 2024). The data collection has been divided into datasets focusing on continuous (netCDF) and discrete (CSV) sampling. Continuous sampling has been subdivided by deployment period. Each of the three deployment periods has a cor-



Earth System Discussions
Science
Data

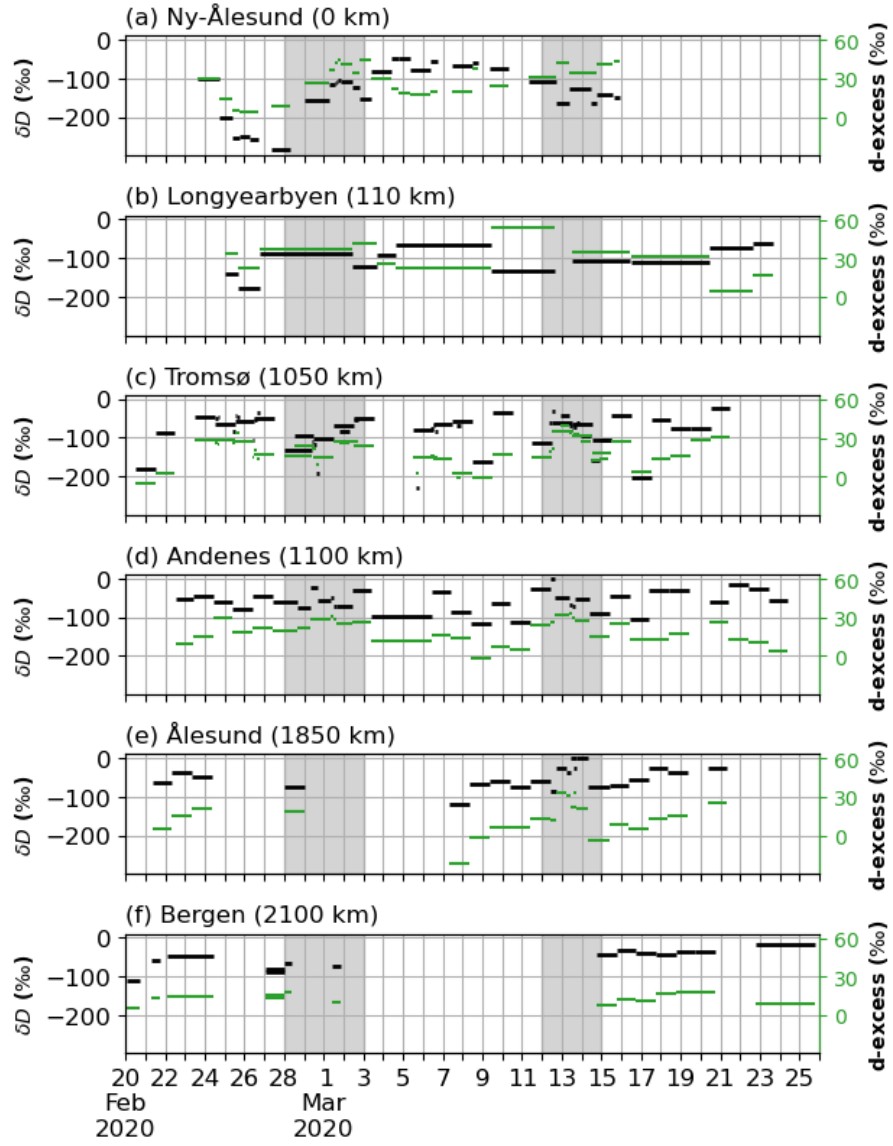

**Figure 8.** Isotopic values of precipitation collected during the campaign. Left axis (black) indicates $\delta$D, whereas right axis (green) indicates the deuterium excess. Locations are as follows, with approximate distance from Ny-Ålesund given in brackets: (a) Ny-Ålesund, (b) Longyearbyen, (c) Tromsø, (d) Andenes, (e) Ålesund, and (f) Bergen. Grey shading indicates the two IOP periods for precipitation sampling.



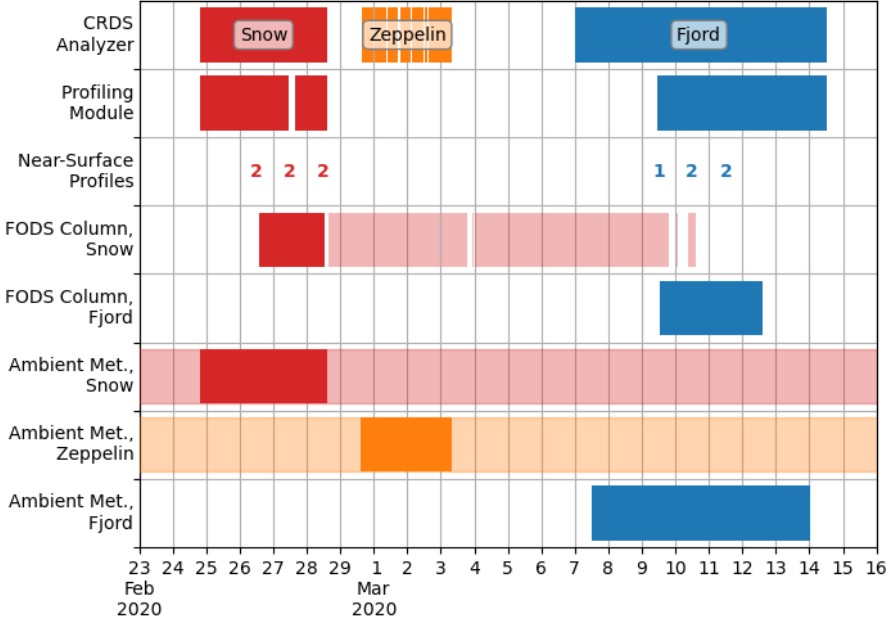

**Figure 9.** Data availability during ISLAS2020 for continuously monitoring instruments during the primary field experiment in Ny-Ålesund, Norway. Dark colored bars indicate periods included in the ISLAS2020 dataset files (Red: ISLAS2020_Snow.nc, Orange: IS-LAS2020_Zeppelin.nc, Blue: ISLAS2020_Ford.nc). "Ambient Met, Fjord" available at raw 20 Hz resolution (ISLAS2020_Fjord_EC.nc). Faded bars indicate data availability beyond the scope of the ISLAS2020 dataset. Number of discrete events given for the calendar day (UTC).

responding netCDF file containing the continuous parameters measured during that time. Missing time intervals at Zeppelin coincide with instrument calibration or the switching of inlets. Missing time intervals at Snow or Fjord coincide with instrument error/failure. The complete 20 Hz dataset from the eddy covariance station is also made available for interested data users. An overview of data availability for the continuously sampled variables at the three deployment sites can be found in Figure 9.

Data from discretely collected water samples is found in a tab-delimited file. In addition to the isotopic analysis for each sample, every discrete sample is associated with metadata relating to its collection. This collection metadata includes the time period, location, and type of water sample. Samples taken in Andenes can also be found as a part of the COMBLE campaign dataset (Sodemann et al., 2020). An overview of data availability for the discretely collected water samples variables for the entire campaign can be found in Figure 10.

Some of the instrumentation included in the ISLAS2020 datasets extend beyond the time period covered in the dataset file, namely the ambient meteorological conditions from the Snow and Zeppelin weather stations, and the full installation period of the Snow FODS column. Corresponding time periods of data availability are indicated with light shaded bars in Figure 9. Principal investigators for these expanded data sources can be found in the metadata of the respective dataset file.
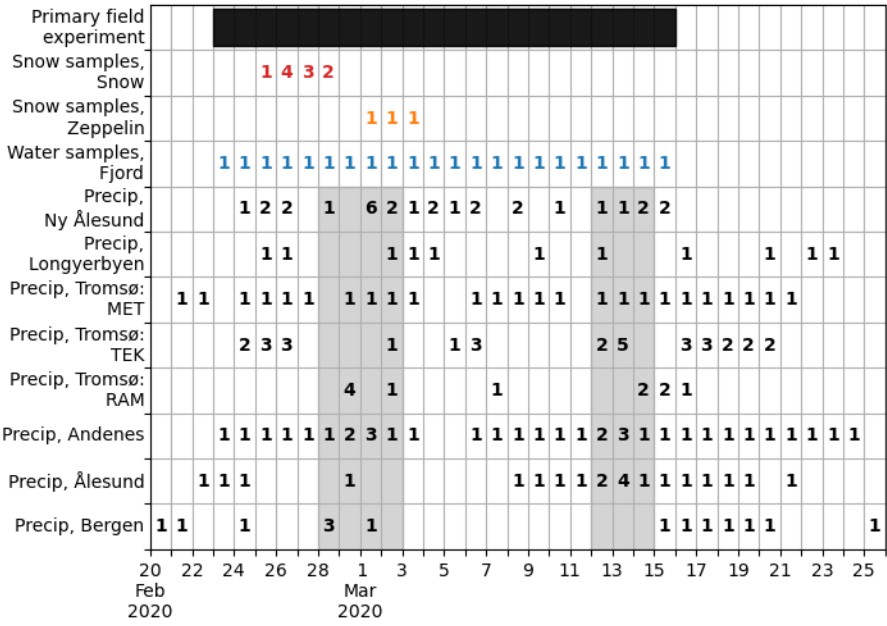

**Figure 10.** Data availability during ISLAS2020 for discretely collected samples included in the ISLAS2020 data collection (IS-LAS2020_Field_sampling.tab). Number of discrete events given for the UTC calendar day, with collection time used to determine day. For reference, the overall duration of the primary field experiment in Ny-Ålesund is indicated by the black bar (time scale of Figure 9). Grey shading indicates the two IOP periods for precipitation sampling.

## 7 Discussion

### 7.1 Measurement uncertainties and limitations

As with other measurements in a challenging field environment, our dataset has sources of uncertainty that can impose limitations for use and interpretation. Foremost, the low humidities encountered during the experiment impart an elevated level of uncertainty to the water vapour isotope measurements. Higher uncertainty stems from the twofold effect of the decreased precision of the analyzer at low humidities and the mixing ratio-isotope dependency attributed to the instrument, which is especially pronounced at low humidities (Aemisegger et al., 2012). The mixing ratio-isotope ratio dependency can be accounted for by a thorough characterization device before and after deployments (Sec. 4.1.1) (Sodemann et al., 2023a) while the lower precision is inherent to the analyzer and can only be mitigated by measurement strategy, such as averaging times. As described in Seidl et al. (2023), each height interval of the profile lasted for a minimum of 4 minutes. Together with the increased flow rate through the measurement cavity of the analyzer, this strategy results in a more robust statistical average for each height level, and therefore a more resolved profile.

In addition, flow distortion around the profiling sites might have impacted surface exchange processes during the experiments. To address this possibility, the profiling equipment used was designed with a minimal volume (Seidl et al., 2023), and





the profiling arm was orientated such that it was directed upstream of the equipment. This approach did appear to work as intended during the Snow deployment, where prevailing southerly winds met the south-westerly pointing arm unimpeded. At

the Fjord site, however, flow distortion proved to be unavoidable. The large concrete pier (orientated NE-SW) sheltered the profiling inlet during south-easterly winds (Figure 7, grey shading), and served as an impassable "back-stop" to the profiling inlet during north-westerly winds, introducing flow convergence and turbulence to the sampling site. As profiling was limited to lower wind speeds, the impact of this flow distortion at the Fjord site is deemed to be limited, but not negligible.

Additional profile measurements at both sites or similar locations would therefore be highly desirable, and more easy to
attain given the experience documented here and elsewhere (Seidl et al., 2023).

Winds and flow distortion are also a potential source of uncertainty for the precipitation samples collected for water isotope analysis. Snow drift and undercatch of snow are well-known problems for precipitation collection in general (Wolff et al., 2015). For water isotope analysis, the undercatch is a less severe problem than the introduction of previously fallen, loose surface snow into the collector. Additionally, the prompt collection of precipitation before collection is another important

factor (Hughes et al., 2021). During the campaign, a consistent sampling protocol and the available additional metadata on wind conditions allows for identification of samples that could be affected by such post-depositional effects.

### 7.2 Opportunities for coordinated analysis

Owing to the nature of the primary experiment site, there are additional complementary datasets from instrumentation present in and around Ny-Ålesund, a select few of which we present here. Firstly, there is the entirety of the horizontal FODS array from

the NYTEFOX experiment, including multiple sonic anemometers (Zeller et al., 2021; Huss et al., 2021). As this horizontal array also gives information on wind speed and direction, there is potential to analyze how fine-scale advection could impact water vapour isotope composition. Such effects can in particular be visible while the profiling arm was stationary for longer periods of time, such as during overnight periods. In addition, insights into the larger vertical structure of the atmosphere is provided by radiosonde launches 4 times daily, alongside remote sensing of temperature, humidity, and winds. Furthermore,

an identical CRDS analyzer was installed at Zeppelin throughout the ISLAS2020 campaign, with both instruments sampling from the same inlet during the first stage of our deployment at Zeppelin (ZEP1). Interested data users are encouraged to contact any of the Principal Investigators for this data collection for more details on these additional data sources.

The ISLAS2020 field experiment took place while the large, international MOSAiC drift campaign took place in the Arctic (Shupe et al., 2022). Since the MOSAiC drift, lasting from October 2019 to September 2020, also involved water isotope

measurements in vapour, precipitation, sea ice and other components (Brunello et al., 2023), there is a potential to find situations where warm airmasses that intruded into the Arctic (Figure 2b) over Ny-Ålesund were later encountered by MOSAiC instrumentation. Vice versa, cold and dry airmasses that left the Arctic during the different CAO periods may have been over the MOSAiC site beforehand (Figure 2c). The precipitation samples from our collection network may be related to both the ISLAS2020 experimental site measurements and MOSAiC experiment in similar ways. In either case, atmospheric transport

modelling using tools such as the FLEXible PARTicle dispersion model (FLEXPART) (Pisso et al., 2019), are needed to deter-

mine the likelihood of connection between different sampling sites. With sufficiently high probability of connection, associated samples can then give information about the transformation of airmasses entering and leaving the Arctic.

## 8    Summary and Conclusions

In this data paper, we describe and summarize the aims, instrumentation, and measurement activities conducted during the ISLAS2020 campaign from 20 Feb to 25 March 2020, with a particular focus on the field experiment at Ny-Ålesund (78.9° N, 11.9° E), Svalbard, Norway from 23 Feb to 15 March. We describe the setting of the campaign and the variable weather conditions encountered during this exceptionally cold spring period. The primary aim of the field campaign/experiment was to acquire near-surface profiles of the vapour isotope composition over different Arctic surface conditions to characterize surface exchange processes of water vapour at these latitudes. To this end, we deployed high-resolution profiling equipment over snow and over the partially ice-covered fjord environment. Our secondary objectives included contextualizing these profiles, both vertically and at a larger horizontal scale. Vertically, the isotopic state of the free troposphere was briefly observed between profiling deployments, while on a regional scale, a wider network of discrete precipitation samples can be used to analyze the field experiment in terms of upstream/downstream measurements. This data paper serves as a reference document for the deployed instrumentation, dataset location, processing and calibration, auxiliary datasets, and data availability. Furthermore, we provide several examples for the intended use of the dataset.

Water vapour isotope measurements at these high latitudes and weather conditions are challenging in several aspects. This includes the low absolute humidity, which increases noise of the CRDS instrumentation, and overall cold conditions, which also requires non-standard approaches to calibration methods. Nonetheless, as demonstrated by the examples, the dataset is of overall entirely sufficient quality and comprehensiveness to allow scientific studies related to exchange processes over snow, sea ice, and open water in extreme, high-latitude environments focused on both process understanding and model development and verification. The additional precipitation and other discrete water samples furthermore allow to place the local water vapour isotope measurements into the context of other measurement activities with respect to the origin and destination of Arctic water vapour and the ensuing transformations as recorded by its water isotope composition, including the MOSAiC campaign.

## Appendix A:  Isotope–mixing ratio correction surface

## Appendix B:  Dataset variable names

*Author contributions.*  Data collection: All authors; Dataset processing: AWS, AD, HS, JMH, CKT, AS, OH; Writing – original draft preparation: AWS, AJ, JMH, HS; Writing – review and editing: All authors



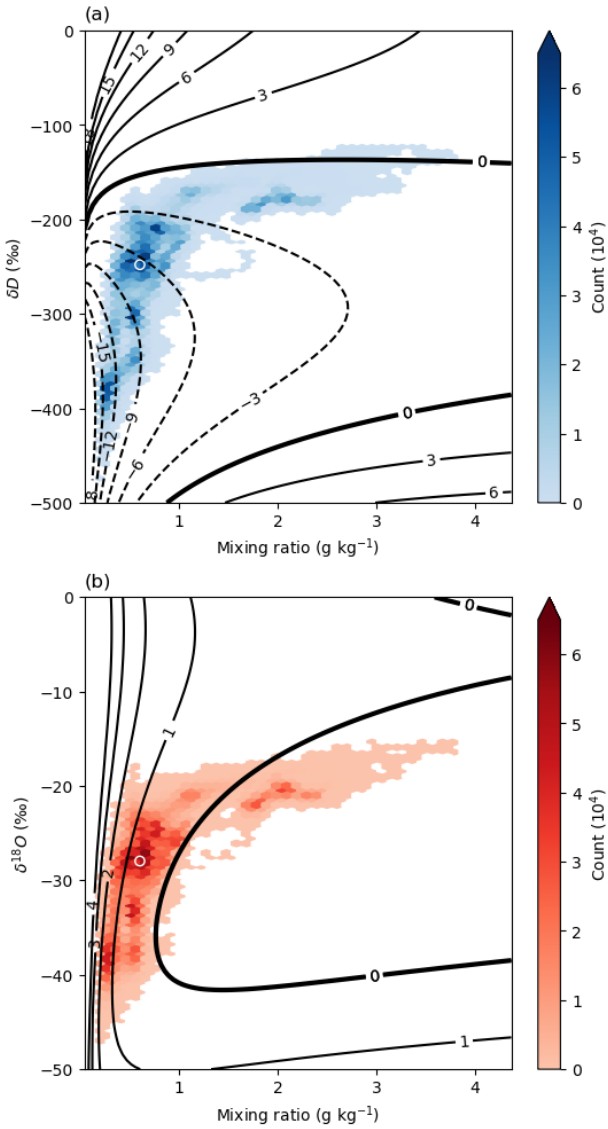

**Figure A1.** Isotope–mixing ratio correction surfaces for $\delta$D (a) and $\delta^{18}$O (b) for the specific Picarro L2130i (Ser#: HIDS2254) used during the ISLAS2020 primary field experiment. Contour values denote correction value needing to be added to raw data. Colored shading denotes concentration of data points from the raw ISLAS2020 dataset. The white circle indicates the median of the dataset for both isotopic value and mixing ratio.





**Table B1.** Names of variables in datasets from Snow (S), Fjord (F), and Zeppelin (Z) deployment sites.

| Grouping | Variable | Dataset(s) | Grouping | Variable | Dataset(s) |
|---|---|---|---|---|---|
| CRDS analyzer | delta_18O | S, F, Z | Meteorological station(s) | Ta | S, F, Z |
| | delta_18O_SD | S, F, Z | | Ta_SD | S, F, Z |
| | delta_18O_LT | S, F, Z | | FF | S, F, Z |
| | delta_18O_LT_SD | S, F, Z | | FF_SD | S, F, Z |
| | delta_D | S, F, Z | | UU | S, F, Z |
| | delta_D_SD | S, F, Z | | UU_SD | S, F, Z |
| | delta_D_LT | S, F, Z | | VV | S, F, Z |
| | delta_D_LT_SD | S, F, Z | | VV_SD | S, F, Z |
| | d | S, F, Z | | p_aws | F, Z |
| | d_SD | S, F, Z | | p_aws_SD | F, Z |
| | d_LT | S, F, Z | | Ta_050_cm | S |
| | d_LT_SD | S, F, Z | | Ta_100_cm | S |
| | q | S, F, Z | | Ta_150_cm | S |
| | q_SD | S, F, Z | | FF_050_cm | S |
| | p | S, F, Z | | FF_100_cm | S |
| | p_SD | S, F, Z | | FF_150_cm | S |
| | Tc | S, F, Z | | UU_050_cm | S |
| | Tc_SD | S, F, Z | | UU_100_cm | S |
| | pc | S, F, Z | | UU_150_cm | S |
| | pc_SD | S, F, Z | | VV_050_cm | S |
| | Twb | S, F, Z | | VV_100_cm | S |
| | Twb_SD | S, F, Z | | VV_150_cm | S |
| | Tdas | S, F, Z | | Tsurf | S |
| | Tdas_SD | S, F, Z | | Tsurf_SD | S |
| | crds_flag | S, F, Z | | WW | F |
| Profiling module / Inlet details | inlet_height | S, F | | WW_SD | F |
| | inlet_height_SD | S, F | | wind_diagnostic | F |
| | profile_number | S, F | | q_aws | Z |
| | Ta_inlet | S, F | | q_aws_SD | Z |
| | Ta_inlet_SD | S, F | | Ta_q_aws_quality | Z |
| | direction | S, F | | FF_UU_VV_quality | Z |
| | subdeployment_number | Z | | p_aws_quality | Z |
| FODS column | Tcol | S, F | Kartverket | z_above_water | F |
| | Tcol_SD | S, F | | z_offset | F |
| | Tcol_inlet | S, F | | max_inlet_height | F |
| | Tcol_inlet_SD | S, F | | | |
| | z | S, F | | | |
| | Tsnow | S | | | |
| | Tsnow_SD | S | | | |





*Acknowledgements.* We would like to thank all of the Kings Bay staff present in Ny-Ålesund during the ISLAS2020 experiment, especially Marine Ilg at the Kings Bay Marine Laboratory. We also want to acknowledge the hard work and dedication of the additional members of the NYTEFOX field experiment: Marie-Louise Zeller, Lena Pfister, Daniela Littmann, and Johann Schneider. The ISLAS2020 experiment also would not have been possible without the fine infrastructure and hosting provided by the Norwegian Polar Institute, especially Christina Pedersen. Similarly, the entire staff at the joint French–German AWIPEV research station in Ny-Ålesund (operated by the Alfred Wegener

Institute for Polar and Marine Research (AWI) and the French Polar Institute Paul-Émile Victor (IPEV)) station must be acknowledged for their support of both the NYTEFOX and ISLAS teams, especially Wilfried Ruhe. Additionally, we wish to thank the Norwegian Institute for Air Research (NILU) for the assistance rendered during our measurements at Zeppelin, especially Dorothea Schulze. We also want to express our gratitude to all individuals involved in the precipitation network collections, including: the UNIS staff, Vitaly Dekhtyarev, the staff at the Vervarslinga for Nord-Norge office, COMBLE field personnel (namely Juarez Viegas), Idar Barstad, and Alexandra Touzeau. We give

hearty thanks to the technical staff at the Geophysical Institute, University of Bergen, who worked long and hard to help prepare equipment and handle logistics for the campaign, especially Anak Bhandari and Helge Bryhni. Finally, we would like to acknowledge the fine work of the data curators at PANGAEA for their expedited efforts to process the associated dataset. This work has been funded by the European Research Council (Consolidator Grants ISLAS (project. 773245) and DarkMix (project. 724629)), as well as by the Research Council of Norway (project no. 291644) Svalbard Integrated Arctic Earth Observing System (SIOS) Knowledge Centre, operational phase.





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
