# Peer review of "The ISLAS2020 field campaign: Studying the near-surface exchange process of stable water isotopes during the arctic wintertime"

_Earth System Science Data, 2024_

## Author Comment (AC1)

Please find our responses to the comments of Referee #1 given below in red text

**Referee #1 Review of Seidl et al. "The ISLAS2020 field campaign: Studying the near-surface exchange process of stable water isotopes during the arctic wintertime" submitted to ESSD (ESSD-2024-293)**

This paper describes an extensive effort of water sample collection for isotope analysis in the Arctic during an exceptionally cold period in late winter 2020 with 11 near-surface profiles over a snow covered and an open ocean surface in Ny Ålesund as well as many precipitation samples from different location in Svalbard and along the coast on the Norwegian mainland. To me the near-surface profile data is truly innovative and interesting and the whole dataset provides exciting new insights into moisture cycling in the Arctic. I find the paper well written and very well documented. In some instances, the descriptions are a bit too detailed and lengthy, but in general I found the paper to be well-balanced. I would suggest providing an earlier overview figure of the sampled data together with the meteorological documentation at the beginning of the paper. Otherwise, the data and methods section is very challenging to read. But this can be easily solved by placing Fig.9 and 10 much earlier and relating them to Fig. 2 with the meteorological overview. I also recommend some more in-depth discussion of the isotope observations in the profiles to provide some physical consistency checks since the deuterium excess observations are indeed far from the normal range of expected measurements. Below you find some more detailed comments.

Thank you. We will re-evaluate the order of figures presented, such that Figures 9 and 10 act as a reference point for the reader through earlier sections. We will also include more references to existing literature to support the plausibility of the negative d-excess observations we observed, with more specific details on this point in the minor comments below.

Minor comments:

- 6: after "primary" one expects to find a "secondary", but was the precipitation sampling campaign really a secondary component of the field experiment? Or just at a larger scale?

The Ny-Ålesund experiment was the "primary" effort of the campaign, in the sense that it was the most focused. The precipitation collecting was indeed not secondary, but the workload was more distributed. This and all following sentences will be rephrased to reflect this.

- 12: meteorological station data?

This will be changed.

- 36: put Pfahl et al. behind regional models and Brady et al behind earth system models.

These references will be moved accordingly.

- 39-40: "as well as the conservation of the isotopic imprint during further airmass transformation over open waters" what do you mean exactly by this? At different stages of the airmass transformation process? Or why "conservation"? Do you expect the isotopic imprint to be conserved during airmass transformation over open waters? I think this is a bit unclearly formulated. Also, to me it seems not so clear how you want to address this question. I really like the profiling aspect, but it's definitely very local and I also like the distributed precipitation sampling, but to me it's not so clear how you want to link them.

We agree that this sentence is unclear. We will modify this and other sentences to better address the distinct knowledge gaps we wish to present, especially in regard investigating airmass transformation as it's transported southward from the High Arctic towards the mid-latitudes.

- 40: I would not write "validate" but evaluate. Otherwise, you start with a biased way of looking at the model assuming it does well in simulating the processes you are interested in.

This change will be made alongside the point above.

- 44: d-xs is a strange notation, I would either opt for d or dexc, or dxs but the "-" is confusing. After introducing it, the abbreviation should be used consistently (e.g. at L. 47 you write d-excess).

Yes, this notation was used in an earlier draft and was not updated as it should have been. Any other occurrences will also be corrected.

- 45-46: You use first non-equilibrium fractionation and then kinetic, that's a bit confusing. Very likely the negative deuterium excess is also due to non-equilibrium fractionation but due to water vapour deposition in supersaturated conditions with respect to surface temperatures (Thurnherr and Aemisegger, 2022). You even write it

yourself later at L. 54 ("freeze dried air masses"). This should be made clearer already here.

This sentence will be reworded in a manner that is more clear, as will any other parts of the manuscript that might unnecessarily confuse the reader on this aspect.

- L 53: Here maybe a reference to AC3 and Kirbus et al. 2024 ACP would be good.

Thank you, we will evaluate adding this reference to the manuscript.

Kirbus, B., Schirmacher, I., Klingebiel, M., Schäfer, M., Ehrlich, A., Slättberg, N., Lucke, J., Moser, M., Müller, H., and Wendisch, M.: Thermodynamic and cloud evolution in a cold-air outbreak during HALO-(AC)[3]: quasi-Lagrangian observations compared to the ERA5 and CARRA reanalyses, Atmos. Chem. Phys., 24, 3883–3904, https://doi.org/10.5194/acp-24-3883-2024, 2024.

- L45: "The HDO molecules are more likely to evaporate than the $H_2^{18}O$ molecules, a process known as kinetic isotope fractionation, which results in a positive d-excess signature" That's not exactly true. HDO molecules are less likely to evaporate than $H_2^{18}O$ molecules. HDO has a lower saturation vapour pressure, and this effect dominates both the $d^{18}O$ and $d^2H$ signals. In equilibrium the $d^{18}O$ and $d^2H$ are related by a factor of about 1 to 8. It's just that there is an excess of HDO in the vapour compared to the equilibrium case during subsaturated conditions.

We will rephrase this sentence to better describe our intended aim, which is as you have described.

- 42: I am not sure I can follow here: I don't know of any theory that would have the ambition to predict the Arctic dexcess surface flux as such. Do you mean that we have an idea about the range of expected dexcess in the evaporation flux for a given range of temperatures and near-surface humidity gradients over the ocean? But there is still ongoing important controversy about the influence of snow metamorphism and what the isotope composition of the net snow sublimation flux is (see Wahl et al. 2024 TC).

Please see the combined response in the point below.

- 42-50: Here I miss a statement that clearly summarises the literature already available on relevant processes. It has been shown in several recent studies (e.g. Thurnherr and Aemisegger, 2022, Brunello et al. 2024 GRL) that snow/ocean-air exchange is the key process that impacts dexcess in the mid- to high latitude marine

boundary layer. The large-scale drivers of this dexcess variability has been linked to warm vs. cold air advection over the mid- to high latitude oceans. What is challenging in the Arctic and around Svalbard is that there are very large inhomogeneities in the surface conditions and both air-sea and air-snow interactions matter. In addition, not only airmass transport and temperature advection play a role such as within the core of the storm track regions, but also prolonged longwave cooling during stable anticyclonic conditions. During these conditions air-snow exchange can be enhanced due to snow metamorphism either within the surface snow (Casado et al. 2021 GRL) or during transport of blowing and drifting snow (Wahl et al. 2024, TC). So, I would say that there are already several literature building blocks available that provide the physical basis for establishing a more nuanced view for reading the Arctic dexcess signals. I suggest merging the paragraphs at L. 42 and L. 51 and discussing the large-scale drivers and physical processes directly with the associated known dexcess signals (near-surface vapour dexcess during upward vs. downward net fluxes, role of snow metamorphism).

These are good points, and similar suggestions were made by Referee#2. We will rework this part of the introduction to better achieve their purposes. Namely, we aim to point out some of the inconsistencies in the literature, and how the datasets described in this paper are primed to give observational evidence to resolve some of these inconsistencies. In this regard, including some of your provided examples from the existing literature as additional "building blocks" would present a more solid foundation for our observations.

- 59: Here the reader needs to know why this reconciliation between lab and field studies is necessary and what it entails. What do lab and field studies not agree upon?

This sentence refers to the mismatch mentioned in L.42-49 and will be rephrased alongside the point above.

- 76: the the

This will be corrected

- Section 2.1: the deployment times don't become clear from this section. How close in time where the three sites visited? Does the free tropospheric site really give a representative observation of the weather situation in which the observations at the two profiling sites were done? Also I didn't find a figure showing these observations neither a discussion of how they could related to the observations at the profiling sites.

The dates covered will be indicated in the text, and Figures 9 and 10 will move closer to this section, thus providing a better orientation for the reader. We will rephrase this paragraph to highlight information about the vertical differences. See Section 7.2

- Section 2.2: it would help to have the abbreviations of the sampling locations of Fig. 2 in the text as well. Furthermore, a timeline with an overview of the sampling periods for the samples taken at the different locations and sublocations would greatly help to get an overview of how much precipitation was sampled where and over which accumulation period. Figure 9 and 10 should be placed here and not in the results section.

These abbreviations are given initially (L.117) but they will also be included in the subsequent paragraphs detailing the specific sites. Figure 10 will also likely move to this section; however, we believe that Figure 9 relates better to Section 3.

- 149: A reference to the climatological work on Fram Strait CAOs and their preconditioning would be helpful here: Papritz et al. 2019

Thank you, we will consider including this reference here.

Papritz, L., Rouges, E., Aemisegger, F., & Wernli, H. (2019). On the thermodynamic preconditioning of Arctic air masses and the role of tropopause polar vortices for cold air outbreaks from Fram Strait. *JGR*, 124, 11033–11050. https://doi.org/10.1029/2019JD030570

- 155: the COAi has units of K or °C I assume.

Indeed, the units should be K. This will be corrected.

- 158: the periods of profiling and where the profiles were done should also be listed clearly in Fig.2: I suggest splitting Fig. 2 into two: one figure covering the timeline in terms of sampling and meteo (as is) and another figure with a synoptic overview (where maybe 2 additional timesteps could be chosen, which are more representative for the profile sampling).

Following the suggestion of promoting Figures 9 and 10 to around this part of the manuscript, we believe that the synoptic context of the profiling periods will be better communicated without adding more subpanels to Figure 2.

- 176: nearby the meteorological station

This sentence will be rephrased to "near to the meteorological station of the observatory."

- Figure 3 and other locations: could the times be indicated in UTC or is there a reason for doing otherwise?

We will ensure that all times given are indicated as "UTC".

- 193-201: This is a very detailed paragraph on the FODS the data of which is published elsewhere, is this really necessary in the text? The deployment dates are relevant for the reader of this paper though.

We will evaluate how this paragraph can be shortened.

- 210: **the** isotopic evolution of the snow

This will be corrected.

- Table 2: times of sampling are missing

We decided to omit specific times, as the days covered are intended as a general overview (e.g. we sampled Fjord every day, even when not profiling, but only surface snow while at the Snow site). We believe this will be even clearer by promoting Figure 10 to Section 2, as suggested previously.

- Table 5: what does this imply for the response time of the instrument?

Unfortunately, characterizing the total inlet response during ZEP1 and ZEP3 was not possible, due to the inaccessibility of the main sampling inlet on the observatory mast. We can only speculate that the response time is dominated by the flowrate through the ~21 m of the observatory inlet manifold. And we consider the response time during ZEP2 to be comparable to the response time at the profiling sites.

- 350: "for a sample at -10 ‰d18O and -100 ‰ is thereby estimated as 0.44 ‰and 1.5 ‰, respectively" space between ‰d18O and missing dD indication.

Thank you, these typos will be corrected.

- 374: at this time resolution the response time of the whole system is a key missing information.

This information is provided in the AMT manuscript presenting the profiling system. This reference will be added to direct the reader.

- 408: double 10 s information

The sentence will be corrected to only mention the time resolution once.

- Section 5.1 I think the profiles are exciting and THE big innovation of this paper: I would find it very valuable to provide the standard profiling plots for all the sampled profiles at the two sites in the supplement. Furthermore, the specific humidity is missing in the profiles, as well as the relative humidity with respect to the surface temperature (key variable to understand air-surface water isotope fluxes) and I would also be curious to see the d18O.

We will be publishing the (Python) code used to generate the profiling period timeseries in an online repository, to be included in the "Code availability" section of the manuscript. With this script it should be possible to visualize any of the dataset variables for a given time period.

- Section 5.1: here the information on the snow and ocean isotope composition would be very important to have together with the profiles. That's why they are useful, namely in combination with the surface profile observations.

Showing surface isotopic composition in Figures 6 and 7 substantially stretches the isotopic scale, and we believe that displaying the equilibrium vapour of the surface (especially at Snow) imparts a level of interpretation that goes beyond the scope of ESSD. However, we will provide the isotopic values of the underlying surfaces during the times covered in Figures 6 and 7 in the text of Section 5.1.

- 450-460: I think for quality check reasons, the low deuterium excess data in air found here should be discussed in terms of their physical plausibility. Clearly the temperature profiles shown for the snow site indicate very strong heat and moisture deposition fluxes to the surface. There is some literature available on this kind of phenomena and their impact on the isotope signature of water vapour in polar regions:

Negative values of the deuterium excess were also found in other studies e.g. during Mosaic and ACE and were associated with warm advection:

Brunello, C.F., Gebhardt, F., Rinke, A., Dütsch, M., Bucci, S., Meyer, H., et al. (2024). Moisture transformation in warm air intrusions into the Arctic: Process attribution with stable water isotopes. *Geophysical Research Letters*, 51, e2024GL111013. https://doi.org/10.1029/2024GL111013

And in the Southern Ocean:

Thurnherr, I. and Aemisegger, F.: Disentangling the impact of air–sea interaction and boundary layer cloud formation on stable water isotope signals in the warm sector of a Southern Ocean cyclone, Atmos. Chem. Phys., 22, 10353–10373, https://doi.org/10.5194/acp-22-10353-2022, 2022.

In their Fig. 2 Thurnherr and Aemisegger, 2022 illustrate and explain the large amplitude change in dvapour compared to d18Ovapour during the process of water vapour deposition to the surface, although over the ocean.
Other studies have discussed negative deuterium excess signals in polar regions as potentially due to sublimation:

Hu, J., Yan, Y., Yeung, L. Y., & Dee, S. G. (2022). Sublimation origin of negative deuterium excess observed in snow and ice samples from McMurdo Dry Valleys and Allan Hills Blue Ice Areas, East Antarctica. *Journal of Geophysical Research: Atmospheres*, 127, e2021JD035950.

Wahl et al. 2024 hint towards the fact that fractionation due to air-snow interactions is likely not due to the sublimation part of the flux but to the depositional part of snow metamorphism (ongoing transformation of the physical structure of the snow important in particular in environments with a strong vertical temperature gradient), and show some evidence for the fact that the dexcess is lowered due to this process during a controlled wind tunnel blowing snow experiment.

Interestingly this profile was sampled during a CAO period over Fram strait. But the profiles with stable stratification for most of the time show that locally the site is influenced by a mesoscale wind system apparently advecting relatively warmer subsiding air over the snow site. I think these aspects should be highlighted because they matter for the credibility of the observations presented, which do deviate somewhat from the normal observational range for the dexcess.

Thank you for this detailed and well-presented point. At the end of the paragraph describing the Snow profiling site (~L.461), we will include a few sentences that direct the reader to some of the existing literature describing possible mechanisms that can produce such low d-excess values. How these mechanisms can help explain in the observed d-excess signals at our measurement profiles is being covered in an upcoming manuscript using these datasets.

- 462-476: Very nice illustration of a situation in which the closure assumption (i.e. that the water vapour isotope signal is the same as the isotope signal of the flux) is far from being satisfied. This could be mentioned here.

We consider including such interpretation beyond the scope of the journal. But we thank you for providing us with this very interesting thought!

- Section 5.1: It is interesting to note that slight variations in the wind direction lead to substantial changes in the vertical temperature structure going from moist plumes over the ocean and subsiding air pockets likely leading to enhanced vapour deposition to more well mixed conditions over the whole column. Therefore, from what I see the temporal variability at one location is at least as large as the vertical variability sampled with the profiling arm. So individual eddies really dominate the temporal variations and the measured isotope signals at different levels. The profiling system is not fast enough to give insights into the vertical structure of one single dynamical feature of vertical transport. This aspect ought to be actively mentioned and discussed.

While the rapid temperature fluctuations contribute to the noisiness of the signal, we conclude that the overall gradient is resolved, since we sampled many heights and analyzed the averages over longer times than the lifespan of individual eddies. But we have also generally thought on how the high-resolution temperature profiles can be used to interpret our isotopic timeseries at a higher frequency; its potential is very exciting.

- Section 5.1: what does a wind direction change from 180° to 250° at the snow site imply for the air mass origin. How comes the vertical column is so differently stratified during these wind direction changes? Is there wind shielding or turbulence induced by the local infrastructure or by people?

In Section 2.1, we discuss the prevailing wind directions and their sources. Figure 1c also shows the topography associated with those directions relative to our profiling sites. And there was no localized wind shielding or turbulence induced upstream.

- [Figures] 6-8: error bars would be very helpful.

Earlier versions of these figures included error shading, however this led to significant complexity in the figures, especially for the isotopic measurements. We will include a representative scaling bar for the variability in the isotopic measurement subplot of Figures 6 and 7.

- Section 5.2: given the results from the previous section surveying conditions at the moisture source and then presenting the precipitation isotopes: I think a short discussion on the importance of the transformation of the signal underway due to fractionation, in particular, related to cloud processing would be helpful to tie the paper together and provide a more coherent storyline to the reader.

We discuss the synergy between our observations of stable isotopes during evaporation and our larger precipitation collection network in Section 7.2. However, we will include a reference to that section at the end of Section 5, so as to not leave the reader with unmet expectations at this point.

- 530: post-depositional modification of the isotope signals has been discussed to be due to snow metamorphism and quantified in several recent studies (e.g. Casado et al. 2021, Aemisegger et al. 2022, Wahl et al. 2024)

We will review the suitability of referencing these manuscripts in our text.

Casado, M., Landais, A., Picard, G., Arnaud, L., Dreossi, G., Stenni, B., & Prié, F. (2021). Water isotopic signature of surface snow metamorphism in Antarctica. *Geophysical Research Letters*, 48, e2021GL093382. https://doi.org/10.1029/2021GL093382

Aemisegger, F., Trachsel, J., Sadowski, Y., Eichler, A., Lehning, M., Avak, S., & Schneebeli, M. (2022). Fingerprints of frontal passages and post-depositional effects in the stable water isotope signal of seasonal Alpine snow. *Journal of Geophysical Research: Atmospheres*, 127, e2022JD037469. https://doi.org/10.1029/2022JD037469

- Good idea, one February 2020 event is described in Brunello et al. 2024.

We will review the suitability of referencing this manuscript in our text, as we take it to be associated with the point given above for L.530.

---

## Author Comment (AC2)

Please find our responses to the comments of Referee #2 given below in red text

**Referee #2 Review of Seidl et al. "The ISLAS2020 field campaign: Studying the near-surface exchange process of stable water isotopes during the arctic wintertime" submitted to ESSD (ESSD-2024-293)**

General comments:

This paper presents water isotope data collected during the ISLAS202 campaign. These stable isotope data include multiple different water phases (e.g., liquid, vapor, solid-snow, ice, etc.) from both inland and coastal settings. Overall, the study design and data collection methods are sound and the data seem of high quality, especially given the difficulties of doing this type of work (the water vapor data, in particular) in the High Arctic. The data presented by the authors is likely to be of use to various different disciplines from climate modelers to cryosphere scientists. The vertical profile data are particularly innovative and of interest. With some minor revisions, this manuscript could be acceptable for publication in Earth System Science Data.

Thank you for your recommendation. We will work to address the minor comments mentioned below.

More specific comments:

- Lines 48-50: Adding some recent (existing) work showing how different process and locations influence variability in Arctic d-excess would be beneficial. While more Arctic d-excess information (as in this paper) would certainly be helpful, recent work reveals some of this nuance that the authors state as needed and should be included. This would also help place the contributions of this work in better context.

  For example:

Wahl, S., Walter, B., Aemisegger, F., Bianchi, L., & Lehning, M. (2024). Identifying airborne snow metamorphism with stable water isotopes. *The Cryosphere*, *18*(9), 4493-4515.

  Indicates how water vapor d-excess can change with varying snow and (Arctic) atmospheric conditions (e.g., temperature, wind, etc.) in a laboratory setting.

Klein, E. S., Baltensperger, A. P., & Welker, J. M. (2024). Complexity of Arctic Ocean water isotope ($\delta 18O$, $\delta 2H$) spatial and temporal patterns revealed with machine learning. *Elementa: Science of the Anthropocene*, *12*(1).

Reveals nuance and new spatial patterns in Arctic d-excess values.

With this set up, the authors can then more specifically describe their new contributions to understanding d-excess variability (some of which begins at Line 59) and place them in better context. For example, the vertical profiles and quite creative and interesting.

A similar point was made by Referee#1, and the two paragraphs consisting of L.42-58 will be rewritten to include more context amongst the existing literature.

- Line 127-128: The authors state that daily samples were taken, entirely of snow. What if there was not any fresh snow? Were samples collected from the existing surface? Was this done in the same spot (after several days of collection, samples would be further down the snow pack and not near the surface)?

We will rewrite this sentence to better describe the sampling schedule observed at the site. Namely that once a day, given that a sufficient amount of fresh snow had fallen, the snow was collected and homogenized, with an aliquot taken for analysis.

- Line 132: In this context, please explain high frequency. Once a day? Twice a day?

We will include the maximum sampling frequency for this location, which was every 3 hours, mostly during IOPs, but also during more localized heavy snowfall events outside the IOP periods

- Line 188: Is the tubing flow path length the same at 4 cm as 200 cm? Due to logistics, I suspect so, but this would be good to clarify. Were the flow rates the same at all heights?

The flow rates and tubing path lengths were the same at all heights, owing to the design of the profiling arm. We will highlight this fact.

- Line 280: Why was the plastic tubing a combination of Bev-A-Line (~4m) and PTFE tubing (~6m)? I don't think this matters for data collection and I understand the challenges of working in the field, but I was just wondering if there was a particular reason for this.

This was the composition of the tubing leading to the semi-permanent CRDS analyser already installed at the observatory. We connected to a (previously capped) tee-junction with a section of our own PTFE tubing.

- Lines 322-324: It looks like with the secondary standards used for water vapor isotope calibration, DI and GSM1, the most depleted (negative) value is -261 ‰ for δD. However, if I am interpreting this correctly, some of the values are far below this (e.g., Figure 6 from the snow tundra site has values below -340). Is there a reason a standard with a lower value was not used? Table 6 lists GLW, which has a lower value, but it appears this was not used for vapor? Is there a reason a standard with a lower value was not used for calibration and how might this impact the values (e.g., potentially greater error with more depleted values)?

This is a good point. Leading up to the campaign period, GSM1 was our most depleted laboratory standard. GLW was only just being introduced to our laboratory at around this time and was unfortunately unavailable to use during the in-field calibrations. This extrapolation to the depleted values observed during the campaign does introduce the potential for increased uncertainty in the calibration to the VSMOW-SLAP scale. However, GLW was used during the characterization of the analyzer's dependency on the isotope – mixing ratio (Sodemann et al., 2023, AMT). We will elaborate in the revised version of the text.

Sodemann, H., Dekhtyareva, A., Fernandez, A., Seidl, A., and Maccali, J.: A flexible device to produce a gas stream with a precisely controlled water vapour mixing ratio and isotope composition based on microdrop dispensing technology, Atmos. Meas. Tech., 16, 5181–5203, https://doi.org/10.5194/amt-16-5181-2023, 2023.

- Also, this is somewhat subjective, but there are many uses of passive voice, which make the paper longer and more difficult to read. For example, in the first sentence of Section 2, the word "being" can be deleted between "site" and "in".

Thank you for pointing this out; we will review our use of active and passive voice. We believe that the proper mixing of the two, as opposed to only one or the other, can lead to a more enjoyable and impartial reading experience. But it can be a difficult balance to achieve.